**Investigating beach erosion related with tsunami sediment transport at Phra Thong Island,**

**Thailand caused by the 2004 Indian Ocean tsunami**

Ryota Masaya [1], Anawat Suppasri [2], Kei Yamashita [2], Fumihiko Imamura [2], Chris Gouramanis [3]

and Natt Leelawat [4, 5]

[1]Civil and Environmental Engineering, Graduate School of Engineering, Tohoku University, 6-6-06

Aoba, Aramaki-Aza, Aoba, Sendai 980-0845, Japan

[2]International Research Institute of Disaster Science, Tohoku University, 468-1 Aoba, Aramaki-Aza,

Aoba, Sendai 980-0845, Japan

[3]Department of Geography, National University of Singapore, 1 Arts Link, Singapore 117570, Singa-

pore

[4]Department of Industrial Engineering, Faculty of Engineering, Chulalongkorn University, Phayathai

Road, Pathumwan, Bangkok 10330, Thailand

[5]Disaster and Risk Management Information Systems Research Group, Chulalongkorn University, Pha-

yathai Road, Pathumwan, Bangkok 10330, Thailand

*Corresponding author.

*E-mail address*: ryota.masaya.r6@dc.tohoku.ac.jp (Ryota Masaya)

**Abstract**

The 2004 Indian Ocean Tsunami and the 2011 Great East Japan earthquake and tsunami caused large-scale topographic changes in coastal areas. Whereas much research has focused on coastlines that have or had large human populations, little focus has been paid on coastlines that have little or no infrastructure. The importance of examining erosional and depositional mechanisms of tsunami events lies in the rapid reorganization that coastlines must undertake immediately after an event. A thorough understanding of the pre-event conditions is paramount to understanding the natural reconstruction of the coastal environment. This study examines the location of sediment erosion and deposition during the 2004 Indian Ocean Tsunami event on the relatively pristine Phra Thong Island, Thailand. Coupled with satellite imagery, we use numerical simulations and sediment transportation models to determine the locations of significant erosion and the areas where much of that sediment was redeposited during the tsunami inundation and backwash processes. Our modelling approach suggests that beaches located in two regions on Phra Thong Island were significantly eroded by the 2004 tsunami, predominantly during the backwash phase of the first and largest wave to strike the island. Although 2004 tsunami deposits are found on the island, we demonstrate that most of the sediment was deposited in the shallow coastal area, facilitating quick recovery of the beach when normal coastal processes resumed.

**1. Introduction**

The 2004 Indian Ocean Tsunami and the 2011 Great East Japan earthquake and tsunami caused large-scale geomorphologic changes in coastal areas during the erosional phases of inflow and outflow (Pari

et al., 2008; Goto et al., 2011a; Tanaka et al., 2011; Haraguchi et al., 2012; Hirao et al., 2012; Udo et
al., 2013; Imai et al., 2015). In each tsunami event, the erosional phases translocated sediments onshore
and offshore,  and primed the coastal zone for rapid (months to decades) recovery (Choowong et al.,
2009; Ali and Narayama, 2015; Udo et al., 2016; Mieda et al., 2017; Koiwa et al., 2018). However, little
information exists to identify real-time sediment dynamics during the erosional and depositional phases
of tsunami events. In particular, erosive phases mobilize sediments into the onshore (e.g. Jankaew et al.
2008; Gouramanis et al. 2017) and offshore environments (e.g. Feldens et al 2009). Following the tsu-
nami event, both offshore environment and coastal environments are primed for natural processes to
resume and redistribute sediments onshore to restore the coastal environment to similar pre-tsunami
configurations.
However, in many regions, such as the area affected by the 2011 tsunami, extensive engineering
interventions (e.g. levee construction and land level raising) are affecting the natural recovery processes
of the coastal zone. In Japan, plans for coastal reconstruction and defenses are typically formulated
shortly after a tsunami, preventing natural recovery processes (Suppasri et al., 2016) and many locations
have not undergone or been allowed to recover naturally (Udo et al., 2016). These political and engi-
neering interventions make it difficult to observe or predict the natural recovery processes of coastal
areas.
Before an understanding of the recovery processes of a tsunami-affected coastal zone can be
achieved, a thorough understanding of the sediment budget must be determined. The relocation of sed-
iments during the main tsunami erosion and deposition phases establishes the pre-recovery or baseline
conditions upon which natural processes can act to facilitate the recovery of the coastal zone. To deter-
mine the locations of sediment deposition during a tsunami event, the sediment transport dynamics
during the tsunami must be defined.
Unfortunately, real-time data from observations has not been possible to establish quantitative esti-
mates of sediment erosion and deposition during a tsunami event, though qualitative spatial patterns of
sediment process (Udo et al., 2016; Yamashita et al., 2016) have been examined through analyze of
video footage. Prior studies have mainly estimated sediment transport dynamics, such as erosion and
sediment deposition through remote sensing (e.g. Fagherazzi and Du 2008, Choowong et al., 2009;
Liew et al., 2010), and sedimentological and stratigraphic analysis (e.g. Paris et al. 2007; Hawkes et al.
2007; Switzer et al. 2012). However, the information obtained regarding the final results of the sediment
transport process is limited. It is difficult to obtain information on where sediment has eroded and de-
posited (e.g. Pham et al., 2018), and whether topographic changes caused by the local sediment runoff
or deposition are the results of action from inflow or backwash (e.g. Choowong et al., 2009; Paris et al
2007; Switzer et al. 2012). This information determines the sediment budget in the system before and
after the tsunami and is therefore important for considering geomorphic recovery.
Numerical simulations using wave dynamics of an area can reproduce spatial-temporal variations of
the sediment mobility and deposition and can effectively model the sediment transport process using
the wave and sediment characteristics of the natural system. In recent years, the numerical modeling of
tsunami sediment transport has been developed (e.g. Takahashi et al., 2000), improved (e.g. Takahashi
et al., 2011; Apotsos et al., 2011a; Li et al., 2013; Morishita and Takahashi, 2014; Yamashita et al.,
2018) and applied in the field (e.g. Gelfenbaun et al., 2007; Takahashi et al., 2008; Apotsos et al., 2011b;
Apotsos et al., 2011c; Gusman et al., 2012; Li et al., 2014; Arimitsu et al., 2017; Yamashita et al., 2017),
and reproducibility has been confirmed by comparison between the calculated and measured values
(e.g. Li et al., 2012; Ranasinghe et al., 2013; Sugawara et al., 2014a; Yamashita et al., 2015; Yamashita
et al., 2016).
An important consideration in the sediment dynamics during catastrophic marine events (e.g. typhoon
and tsunami) is the degree of development and human modification of the coastal zone prior to the
event. Artificial structures, such as sea walls, roads and buildings interfere with washover processes,
and these areas are often targeted from reconstruction and rehabilitation through rapid engineering re-
construction. Little is known about the recovery processes in sparsely developed and unpopulated areas.
As such, the largely, anthropogenically-undisturbed Phra Thong Island, western Thailand, is an ideal
location to model the sediment dynamics, coastal erosion and deposition following a major tsunami
event.
The main objective of this study is to investigate the short-term conditions of sediment transport such
as erosional and depositional process and establishes the baseline sediment conditions that led to further
investigation of the long-term recovery of the Phra Thong Island coastline after the 2004 IOT. We used
tsunami sediment transport calculations to spatio-temporally reproduce the sediment transport pro-
cesses occurring during the tsunami and identify zones of sediment deposition in the offshore and on-
shore areas and validate these modelling results with published observational data of the 2004 IOT
deposits on the island. Due to the largely natural environment, Phra Thong Island is a rare case that is
useful for verifying tsunami sediment transport models where few artificial features can generate model
uncertainties.
Examining the sediment transport processes on Phra Thong Island is also expected to elucidate phe-
nomena, improve numerical calculation models for the future and is applicable to other areas. Further-
more, at least three palaeotsunami deposits were identified in areas impacted by the 2004 IOT on Phra
Thong Island (Jankaew et al., 2008; Sawai et al., 2009; Fujino et al., 2009; Fujino et al., 2010; Prender-
gast et al. 2012; Brill et al., 2012a, b; Gouramanis et al., 2017; Pham et al., 2018). Thus, clarifying the
sediment transport conditions of the 2004 tsunami will also be important for future estimations of his-
tory, scope and cause of older tsunamis on Phra Thong Island and elsewhere in the coastal areas of the
Indian Ocean.

**2.  Setting and method**
**2.1.  *Phra Thong Island, Thailand***
During the 2004 IOT, a wave of approximately 7 m inundated the northern portion of Phra Thong Island
(Fig. 1) and measurements up to 20 m were recorded from the southernmost tip of the island (Jankaew
et al. 2008). Over 70 people were lost and a village of 100 households disappeared. Geomorphologically,

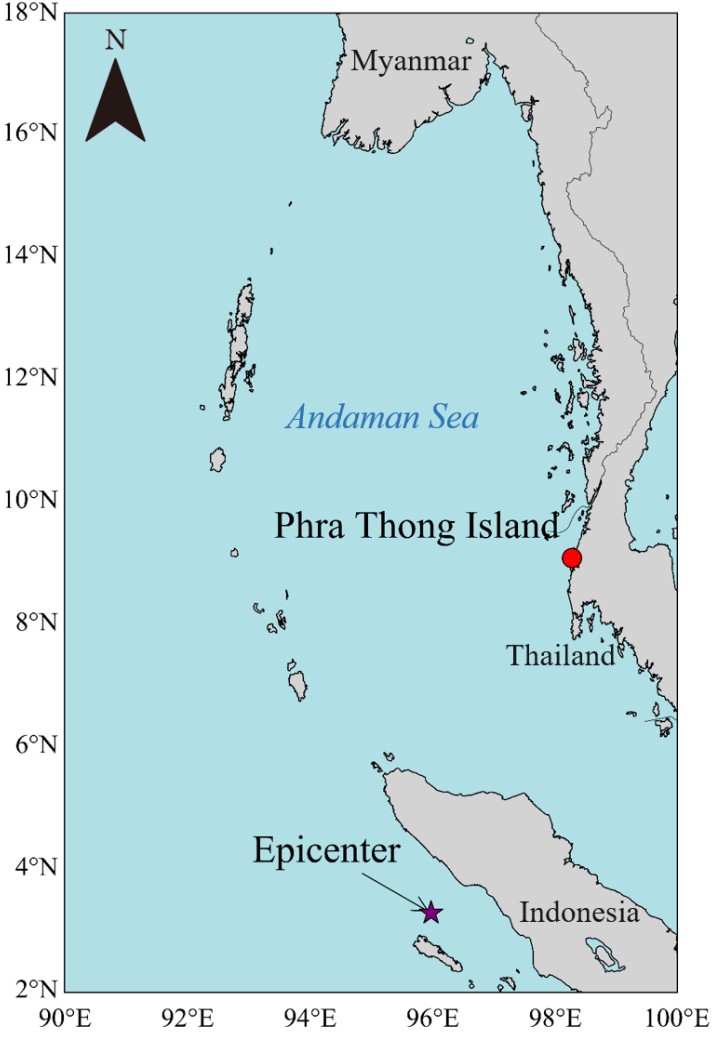

Figure 1 Location of Phra Thong Island

the western coast of the island has a beach ridge sequence trending parallel to the coast, which formed
during the sea level regression following mid-Holocene sea level highstand about 6,000 B.P. (Brill et
al. 2015). The eastern shore of the island is extensively covered by mangroves along the shores of tidal
channels. The island has a tropical climate. Additionally, palaeotsunami deposits are preserved in swales
in the beach ridge system along the western coast of Thailand (e.g. Jankaew et al. 2008; Gouramanis et
al. 2017). Furthermore, although local beaches were lost in the 2004 tsunami, satellite photography
showed rapid natural recovery within 18 months (e.g. Choowong et al. 2009).

### 2.2. *Topography and bathymetry data*

The topography and bathymetry data used for the tsunami sediment transport calculations were cre-
ated based on various water depths and elevations. Figure 2 shows the terrain data that were created.
Topographic data were downscaled from Region 1, which includes the Andaman Sea, to Region 6,
which includes all of Phra Thong Island. The grid spacing decreases from Region 1 (the spatial grid
size $\Delta x_1 = 1{,}215$ m) to Region 6 ($\Delta x_6 = 5$ m). In the tsunami sediment transport calculations, UTM zone

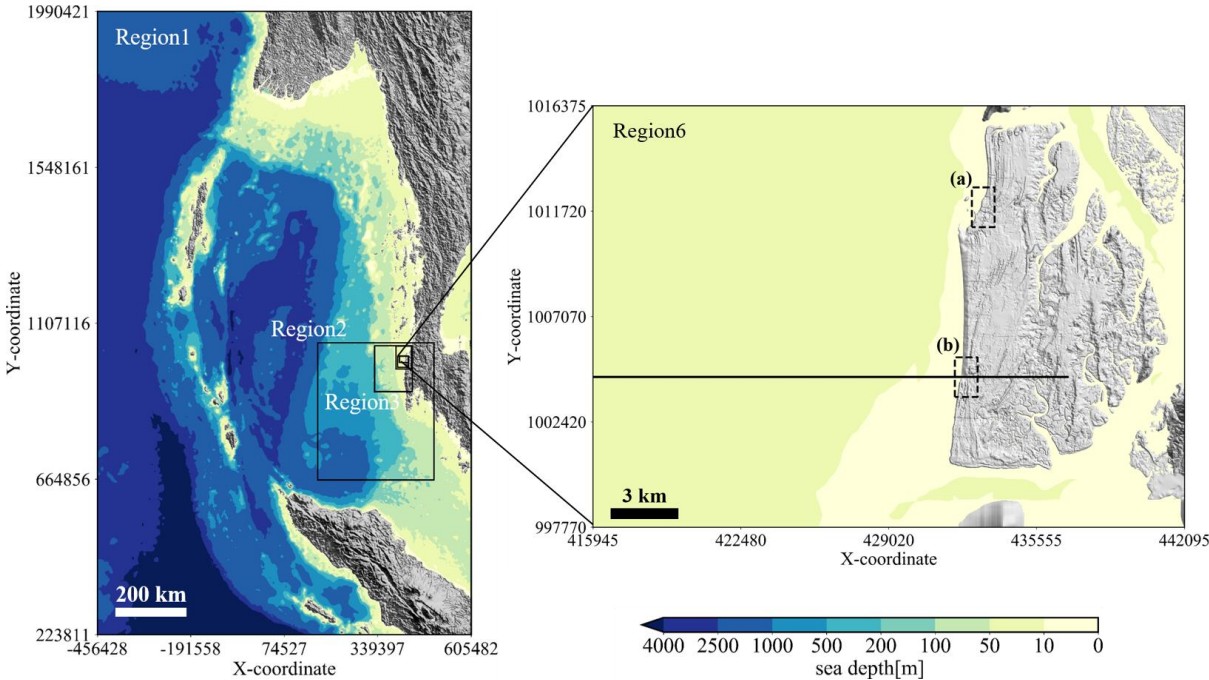

Figure 2 Terrain data (The black frame shows Region 1 to Region 6, and the black line in Region 6 shows the cross-section where calculation was performed. Dashed squares are the beach where erosion was confirmed from satellite image.)

47N was used to geospatially constrain the horizontal modelling coordinates of Phra Thong Island. Region 1 is the projection of depth data of the 30-second grid provided by GEBCO (2014) on the Cartesian coordinate system UTM 47N. Regions 2–4 use a digital marine chart with 300 m resolution based on a survey by the Royal Thai Navy. Regions 5 and 6 use an original 5 m (terrain data) and 15 m (sea depth data) grid spacing to create mean terrain and water depth data based on analysis of satellite image by EOMAP and elevation data provided by the Land Development Department of Thailand (LDD, 2018). The terrain data of Region 4, created from the digital marine chart of 300 m resolution, showed discontinuity at the boundary with Region 5, which had a higher resolution. The discontinuity was therefore removed to the extent possible by interpolation with an inverse distance weighting method using all terrain data.

### 2.3. *Tsunami source model*

The tsunami source model proposed by Suppasri et al. (2011) was used as the tsunami source of the 2004 Indian Ocean Tsunami as the model focused on the coast of Thailand and accurately reproduced the inundation area and surveyed trace height of the 2004 IOT. The fault model is divided into six small faults from satellite image analysis and survey results, and it is assumed that each small fault slides simultaneously and instantaneously. For the tsunami source, the vertical tectonic displacement in each fault was calculated according to Okada (1985). Table 1 shows the fault parameters of each fault and Figure 3 shows the initial water level.

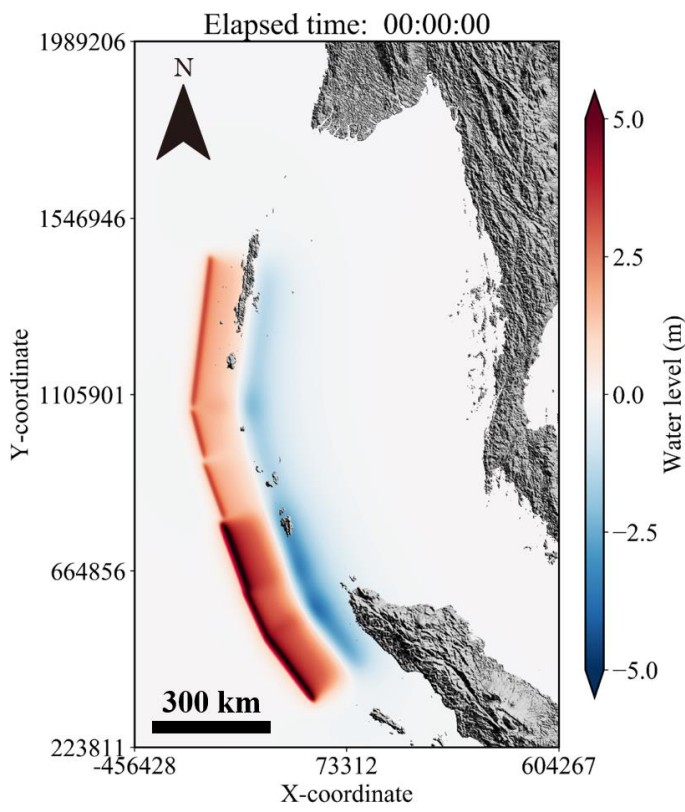

Figure 3 Initial water level after earthquake occurrence

Table 1 Earthquake fault parameters for calculating initial water level (Suppasri et al., 2011)

| Segment No. | 1 | 2 | 3 | 4 | 5 | 6 |
|---|---|---|---|---|---|---|
| Latitude (˚N) | 3.03 | 4.48 | 5.51 | 7.14 | 8.47 | 9.63 |
| Longitude (˚E) | 94.4 | 93.3 | 92.9 | 92.3 | 91.9 | 91.6 |
| Strike (deg) | 323 | 335 | 340 | 340 | 345 | 7.00 |
| Dip (deg) | 15.0 | 15.0 | 15.0 | 15.0 | 15.0 | 15.0 |
| Rake (deg) | 90.0 | 90.0 | 90.0 | 90.0 | 90.0 | 90.0 |
| Length (km) | 200 | 125 | 180 | 145 | 125 | 380 |
| Width (km) | 150 | 150 | 150 | 150 | 150 | 150 |
| Slip (m) | 14.0 | 12.6 | 15.1 | 7.00 | 7.00 | 7.00 |
| Depth (km) | 10.0 | 10.0 | 10.0 | 10.0 | 10.0 | 10.0 |

**2.4.  *Tsunami sediment transport calculation***

**2.4.1.  *Tsunami propagation and run-up model***

Tohoku University's Numerical Analysis Model for Investigation of Near-field tsunamis, No. 2 (TUNAMI-N2) is based on the nonlinear long wave theory and was used as the tsunami propagation and run-up model (Imamura, 1996).

$$\frac{\partial \eta}{\partial t} + \frac{\partial M}{\partial x} + \frac{\partial N}{\partial y} = 0 \tag{1}$$

$$\frac{\partial M}{\partial t} + \frac{\partial}{\partial x}\left(\frac{M^2}{D}\right) + \frac{\partial}{\partial y}\left(\frac{MN}{D}\right) + gD\frac{\delta\eta}{\delta x} + \frac{gn^2}{D^{\frac{7}{3}}}M\sqrt{M^2 + N^2} = 0 \tag{2}$$

$$\frac{\partial N}{\partial t} + \frac{\partial}{\partial x}\left(\frac{MN}{D}\right) + \frac{\partial}{\partial y}\left(\frac{N^2}{D}\right) + gD\frac{\delta\eta}{\delta y} + \frac{gn^2}{D^{\frac{7}{3}}}N\sqrt{M^2 + N^2} = 0 \tag{3}$$

Here, $\eta$ is the change in water level from the still-water surface, $D$ is the total water depth from the bottom to the water surface, and $g$ is the acceleration of gravity. The bottom friction is expressed according to the Manning formula, where $n$ is Manning's roughness coefficient ($n = 0.025$ s m$^{-1/3}$). $M$ and $N$ are the total flow fluxes in the $x$ and $y$ directions, respectively, and are given by integrating the horizontal flow velocity $u$, $v$ from the water bottom $h$ to the water surface $\eta$. It is assumed that the horizontal flow velocity is uniformly distributed in the vertical direction.

The nonlinear long wave theory consists of a continuous equation that is derived from (1) the principle of conservation of mass (continuity equation) and (2) the conservation of momentum (equation of motion). These two equations are obtained by vertically integration from the seabed to the water surface.

When the water depth is about 50 m or less, the effects of the second, third and fifth terms of the advection and seabed friction terms (Equations 2 and 3) are reduced, therefore wave theory that omit these terms is often used at depths shallower than 50 m. Meanwhile, the Message Passing Interface (MPI) parallel was implemented in the model for highly efficient calculations. Both the advection term and the bottom friction term were therefore considered in the calculations without reducing accuracy in deeper waters.

The reproducibility of the calculated results is based on the tsunami height data (IUGG; available at http://www.nda.ac.jp/~fujima/TMD/index.html) for the 2004 IOT and is discussed using the geometric mean $K$ and geometric standard deviation $\kappa$ proposed by Aida (1978).

$$\log K = \frac{1}{n_{\mathrm{p}}}\sum_{i=1}^{n}\log K_i \tag{4}$$

$$\log\kappa = \sqrt{\frac{1}{n_{\mathrm{p}}}\left\{\sum_{i=1}^{n}(\log K_i)^2 - n_{\mathrm{p}}(\log K)^2\right\}} \tag{5}$$

Here, $n_{\mathrm{p}}$ is the number of points, $R_i$ is the tsunami height at the $i$th point, $H_i$ is the calculated value at

the $i$th point, and $K_i = R_i/H_i$.

### 2.4.2. Sediment transport model

For the tsunami movable bed model, we used the numerical Sediment Transport Model (STM) pro-
posed by Takahashi et al. (2000), which solves the time evolution of sediment transport considering the
exchange sediment volume of the bed and suspended load layers according to the flow conditions of
the nonlinear long wave theory-based TUNAMI-N2 model. For each time step in the finest calculation
region (region 6), the STM receives the total flow fluxes from TUNAMI-N2 and calculates the change
of seafloor and land surface and feeds this to the next time step of the TUNAMI-N2 model.
In this model, tsunami sediment transportation is derived into two layers, bed load layer and sus-
pended load layer. In the bed layer, the sediment particles are transported through rolling, sliding or
saltating. In the suspended load layer, the sediment particles are uplifted and transported by the dynamic
of flowing suspension. The governing equations consist of continuous equations for the bed load layer
and the suspended load layer:

$$\frac{\partial Z_B}{\partial t} + \frac{1}{1-\lambda}\left(\frac{\partial q_{B,x}}{\partial x} + \frac{\partial q_{B,y}}{\partial y} + w_{ex}\right) = 0 \tag{6}$$


$$\frac{\partial Ch_s}{\partial t} + \frac{\partial CM}{\partial x} + \frac{\partial CN}{\partial y} - w_{ex} = 0 \tag{7}$$


Here, $\lambda$ is the porosity of the sand particles, $Z_B$ is the bottom height from the reference plane, $q_B$ is the
amount of bed load sediment, $C$ is the average suspended load layer concentration, $h_s$ is the suspended
load layer thickness (= total water depth), and $M$, $N$ are the water discharges in the $x$ direction and $y$
direction, $w_0$ is the settling velocity of the sand particles.
Equation (6) is a continuous equation for within the bed load layer. The first term is the exchange
sediment volume with the bottom, the second term is the balance of sediment flow volume moving in
a tractive form in the flow direction, and the third term defines the balance of suspension flux, caused
by diffusion, and sedimentation flux, caused by gravity, as the exchange sediment volume between the
bed load layer and the suspended load layer.
Equation (7) is a continuous equation for within the suspended load layer. The first and second terms
are bed load sediment moving in a suspended state in the flow direction, the third term is the exchange
sediment volume between the bed load layer and the suspended load layer, and the fourth term is the
increase or decrease of the sediment flow in the suspended load layer.
In Equations (8) and (9), the equations defining the bed load sediment volume $q_B$ and the equation
defining the exchange sediment volume $w_{ex}$ of the bed load layer and suspended load layer are neces-
sary, but according to Takahashi et al. (2000), they are obtained by the following:

$$q_{B,x\,y} = \alpha\sqrt{sgd^3}(\tau^* - \tau^*_{crit})^{\frac{3}{2}} \tag{8}$$

$$w_{ex} = \beta\sqrt{sgd}(\tau^* - \tau^*_{crit})^2 - w_0 C \tag{9}$$

$$\tau_* = \frac{u^{*2}}{sgd} \tag{10}$$

Here, $\alpha$ is the coefficient of the bed load sediment volume equation, $\beta$ is the coefficient of the suspension volume equation, $s$ is the submerged density of the sand particles ($s = \rho_s/\rho_w - 1$; $\rho_s$ and $\rho_w$ are the density of sand particles and water, respectively; $\rho_s = 2{,}650$ kg m$^{-3}$ and $\rho_w = 1{,}000$ kg m$^{-3}$, $g$ is the acceleration of gravity, $d$ is the grain diameter, $w_0$ is the settling velocity of the sediment grains, $\tau^*$ is the Shields parameter (Equation (10)), $\tau^*_{crit}$ is the critical Shields parameter, $u^*$ is the friction velocity obtained from the Manning's law ($u^{*2} = gn_s^2 M|M|/D^{7/3}$). $n_s$ is the roughness coefficient for STM. It is worthwhile to mention that in the tsunami calculation, roughness coefficient $n$ is the parameter to calculate the shear stress from ground surface which is differs from $n_s$, the parameter for calculating the shear stress which forces on the surface of sand particles.

The grain-size dependent parameter for bed load ($\alpha$) and exchange rate ($\beta$) in Equation (8) and (9) are derived from Equations (11) and (12) based on the hydraulic experiments by Takahashi et al. (2011):

$$\alpha = 9.8044e^{-3.366d} \tag{11}$$

$$\beta = 0.0002e^{-6.5362d} \tag{12}$$

However, the functions should not be applied when $d$ is outside the 0.166 mm to 0.394 mm range as he validity of extrapolated $d$ values may produce erroneous results.

$$w_0 = \sqrt{sgd}\left(\sqrt{\frac{2}{3} + \frac{36v^2}{sgd^3}} - \sqrt{\frac{36v^2}{sgd^3}}\right) \tag{13}$$

Equation (13) is a settling velocity of the sand particles by Rubey (1933). Here, $v$ is the kinematic viscosity coefficient ($v = 1.39 \times 10^{-6}$ m$^2$ s$^{-1}$).

Considering the effect from the bed slope (Watanabe et al., 1984), the formulation of the bed load, $q_B$, Equation (6), is rewritten as $Q_B$ as shown in Equations (14) and (15):

$$Q_{B,x} = q_{B,x} - |q_{B,x}|\varepsilon\frac{\partial Z_B}{\partial x} \tag{14}$$

$$Q_{\mathrm{B},y} = q_{\mathrm{B},y} - |q_{\mathrm{B},y}|\varepsilon\frac{\partial Z_{\mathrm{B}}}{\partial y} \qquad (15)$$

where $\varepsilon$ is the parameter which related to the diffusion coefficient of the sediments ($\varepsilon = 2.5$; Sugawara et al., 2014a).

Sediment transport during tsunami largely occurs as suspension (Takahashi et al., 2000). In such situations, suspended sediments are maintained in the water column by turbulence while the energy of the turbulence is dissipated due to the increased suspended sediment concentration. This induced an equilibrium state in which no further sediment supply from the bottom occurs. The resulting concentration is called the saturated sediment concentration. The expression for saturation concentration of suspended sediments $C_{\mathrm{s}}$ is applied as (Van Rijn, 2007; Sugawara et al., 2014a):

$$C_{\mathrm{s}} = \frac{\rho_{\mathrm{w}}}{\rho_{\mathrm{s}} - \rho_{\mathrm{w}}}\frac{e_{\mathrm{s}}n_{\mathrm{s}}^{2}u^{3}}{h_{\mathrm{s}}^{4/3}w_{0} - e_{\mathrm{s}}n_{\mathrm{s}}^{2}u^{3}} \qquad (16)$$

where $e_{\mathrm{s}}$ is the efficiency coefficient ($e_{\mathrm{s}} = 0.025$; Bagnold, 1966). Note that in the sediment transport calculation, the saturation concentration of suspend sediments given by Equation (16) is applied entrainment of sediment from the bottom layer. Namely, sediment supply from the bottom to the water column (suspended load layer) by $w_{\mathrm{ex}}$ is not permitted if $C \geq C_{\mathrm{s}}$. However, supersaturation ($C \geq C_{\mathrm{s}}$) due to sediment advection, or sudden decrease of $C_{\mathrm{s}}$ due to the change of flow parameters, is permitted. In this calculation, when $C$ exceeds maximum concentration $C_{\mathrm{max}}$ was set to 37.7%, based on the observed value (Xu, 1999a, 1999b).

In Equation (6) and Equation (7), the bottom height ($Z_{\mathrm{B}}$) is determined from the reference plane, and the average suspended sediment concentration ($C$) are the initial values before the tsunami and the flow flux ($M$). Because suspended sediment thickness ($h_{\mathrm{s}}$) is given by the equation of motion of a fluid and the continuous equation, sea level fluctuation can be determined over time. Further, the MPI parallel was implemented to enable relatively efficient wide area calculations (e.g. Yamashita et al. 2016).

### 2.5. *Calculation conditions*

The initial conditions for the numerical simulations used the terrain data (Figure 2) and tsunami source (Figure 3). The simulations were performed using a 3:1 nested grid that increased the resolution a 1,215 m² grid to a 5 m² grid. Additionally, the target region of the sediment transport calculation was limited to Region 6, with a grid spacing of 5 m².

The simulations were calculated over a 0.05 second increment with a 6-hour period in which the test case with a 12-hour period showed the suspended sediment concentration in the vicinity of the shoreline decreased and stabilized. Therefore, 6-hour simulation was used for the reproduction of the 2004 tsunami as well as further sensitivity analysis of the grain size and roughness coefficient.

Table 2 Set parameters for sediment transport calculations

| Parameter | Value |
| --- | --- |
| Coefficient of bed load sediment volume equation $\alpha$ | 6.40 |
| Coefficient of suspension sediment volume equation $\beta$ | $8.70 \times 10^{-5}$ |
| Critical friction velocity $u^*_{crit}$ | 0.0137 m s$^{-1}$ |
| Settling velocity of sand particles $w_0$ | 0.00971 m s$^{-1}$ |

For the bottom conditions of STM, the roughness coefficient was fixed at $n_s = 0.030$ s m$^{-1/3}$, and the entire area of Region 6 was considered the movable bed. In general, when simulating tsunami sediment transport, it is necessary to determine the roughness coefficient according to land use. However, since there is no land use map before the tsunami on Phra Thong Island, a fixed value was used, similar to previous studies (e.g. Sugawara et al., 2014a, b; Yamashita et al., 2015; Yamashita et al., 2016). However, Sugawara et al. (2014a) showed that the variation in Manning's roughness coefficient for the sand beds may affect the general distribution pattern of sediment deposits and erosions across the artificial topographic features with much higher roughness coefficient such as artificial canals, roads and populated residential areas. Therefore, a sensitivity analysis on the roughness coefficient was performed. Phra Thong Island has no such artificial topographic features and using the single roughness coefficient should sufficiently capture the overall roughness. However, to ensure robust conclusions, a sensitivity analysis for two bottom conditions was performed at $n_s = 0.025$ s m$^{-1/3}$ and $n_s = 0.035$ s m$^{-1/3}$, which are within the range of previously used estimates of roughness (e.g. Sugawara et al., 2014a, b).

The grain size was based on one sediment data set (Gouramanis et al., 2017) from the locally eroded region, and was considered as a representative value for all of the tsunami sediment grain sizes. A uniform grain size of $d = 0.127$ mm was used. The critical Shields parameter $\tau^*_{crit}$ in Equations (9) and (10) was obtained using Equations (17) and (18) according to Iwagaki et al. (1956):

$$\tau^*_{crit} = u^{*\,2}_{crit}\rho \tag{17}$$

$$u^{*\,2}_{crit} = 8.41 d^{\frac{11}{32}} \tag{18}$$

Here, $u^*_{crit}$ is the critical friction velocity and $\rho$ is the density of water. Table 2 shows each parameter used for the sediment transport calculations in this study.

The numerical model used in this paper can only consider a single grain size, so the model cannot resolve the grading observed in the sand layers (e.g. Gouramanis et al., 2017). Additionally, initial bed grain size can have a large effect on erosion and deposition (e.g. Apotsos et al., 2011b; Sugawara et al., 2014a; Jaffe et al., 2016). Furthermore, the sediment data we used to set the grain size is from a single location in the north of the island, and is assumed to be a representative grain size for the tsunami deposits. As such we performed a sensitivity analysis on the grain size. Pham et al. (2018) investigated

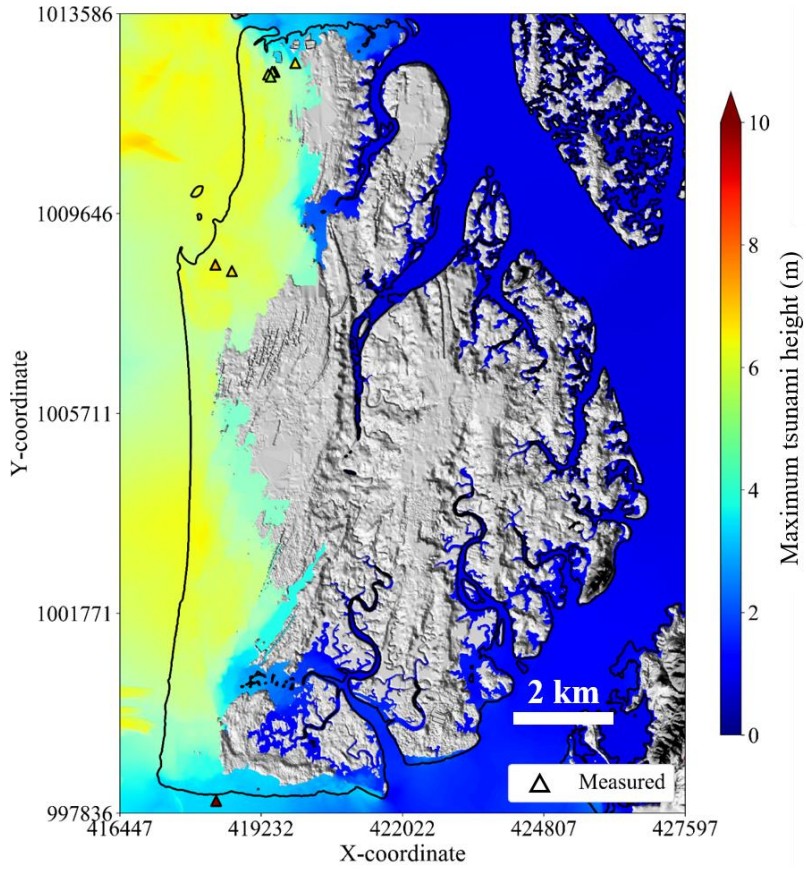

Figure 4 Comparison of calculated and measured maximum tsunami height

the surface grain size of the offshore (water depth > 15 m), nearshore (water depth < 15 m), and onshore on Phra Thong Island, which they considered to be the source of sediments that formed the tsunami deposits. Pham et al. (2018) recorded a mean grain size of 0.314 mm in the offshore area, 0.129 mm in the nearshore area, and 0.285 mm in the onshore area. Based on these mean grain sizes, we conducted a sensitivity analysis for two grain sizes (0.285 mm and 0.314 mm representing the offshore and onshore sediments).

## 3. Results

### 3.1. *Verification of reproducibility*

#### 3.1.1. *Tsunami height*

Figure 4 shows the results of the calculation of the maximum tsunami heights and the seven measured tsunami heights on Phra Thong Island. From Equations (4) and (5), $K = 1.16$ and $\kappa = 1.40$ are obtained. The Japan Society of Civil Engineers (2012) consider $0.95 < K < 1.05$ and $\kappa < 1.45$ as guides for evaluating reproducibility of tsunami numerical calculations. Although the $K$ value is slightly higher than the guideline but this is because of an uncertain 19.6 m measured in the southern part of the Island. Additionally, the source model used in this calculation gives $K = 0.84$ and $\kappa = 1.30$ for reproducibility of tsunami height in the wide area along the coast of Thailand (Suppasri et al., 2011). Therefore, it can be said that this calculation has the same tsunami reproducibility as the previous study.

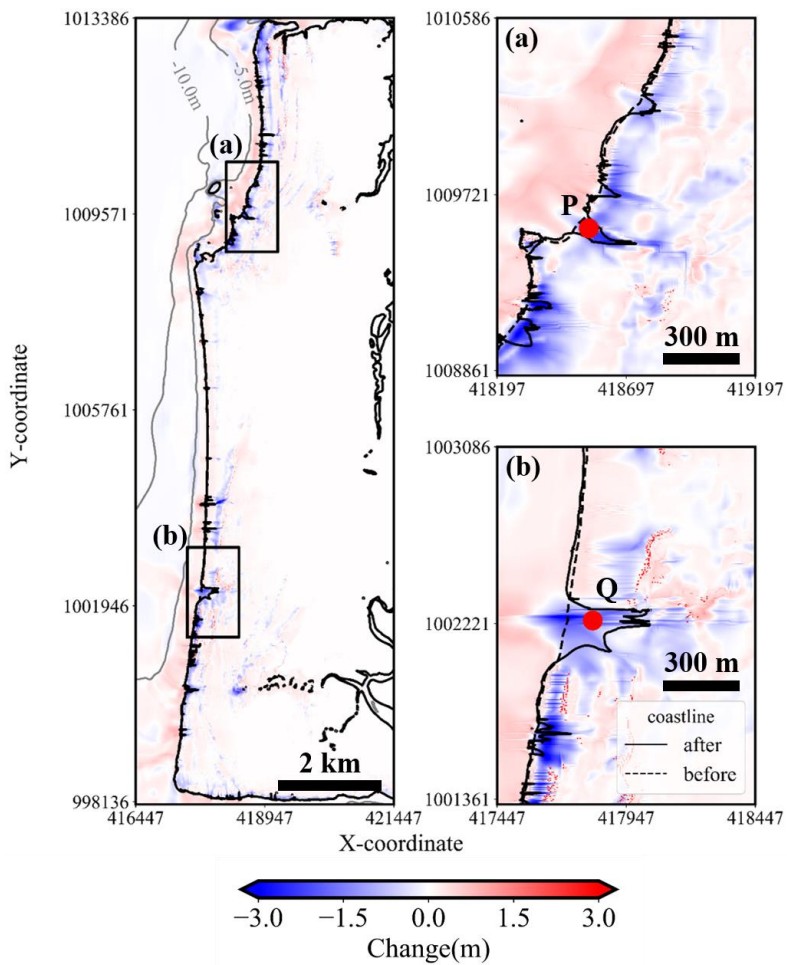

Figure 5 Topographic change and shoreline position caused by the tsunami (solid & dashed lines show that the coastline after and before the tsunami in the simulation, P and Q are the points confirmed local beach erosion in region (a) and (b), blue and red mapping show erosion and deposition after the tsunami in the simulation)

### 3.1.2. *Shoreline changes*

Our sediment transport models identify the locations of significant sediment erosion, which are confirmed from post-tsunami satellite images. Figure 5 shows the pre-2004 IOT topographical and geomorphological features (dashed line) and the modelled changes caused by the tsunami (solid line). Erosion typically occurs locally where small tidal channels breach the youngest beach ridge system (Figure 5(a) and 5(b)). Comparison with the satellite image shows that the position of erosion in both regions is consistent (Figure 6). Although the actual amount of erosion is unknown, this indicates that the planar spread of the eroded component can be well reproduced by the calculation. Region (a) was further investigated in detail, as the area corresponds to the point where sediment outflow occurred (Jankaew et al., 2008).

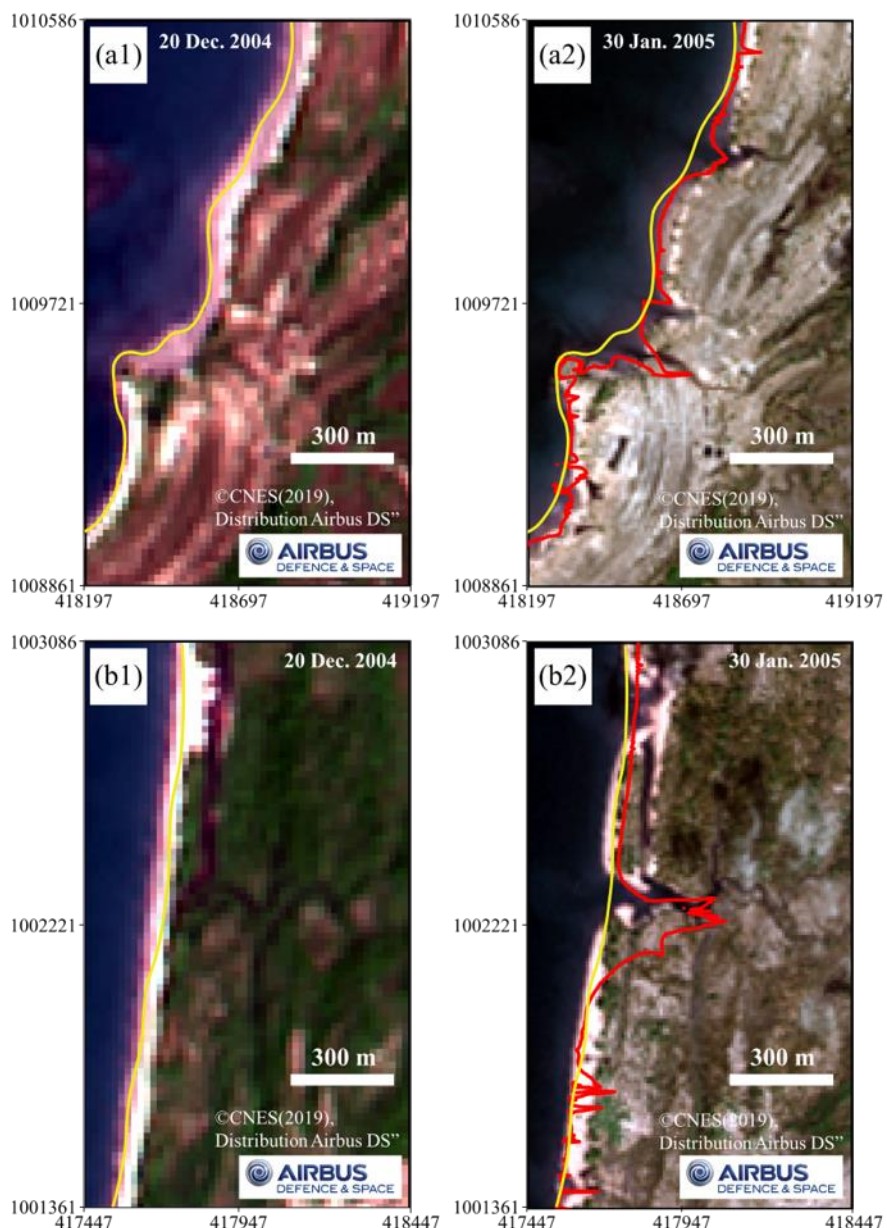

Figure 6 Comparison of observed shoreline position from Figure 5 region (a) and (b) derived from
satellite images before and after the tsunami (20 Dec. 2004 and 30 Jan. 2005), which is overlain by
the modelled extent of erosion showing that the modelled results closely match the observed
changes. The red line is the calculated shoreline after the tsunami, and the yellow line is the shore-
line before the tsunami (© CNES, 2019, Distribution Airbus DS").
a1) Satellite image before the tsunami in region (a), a2) Satellite image after the tsunami in region (a),
b1) Satellite image before the tsunami in region (b), b2) Satellite image after the tsunami in region
(b)

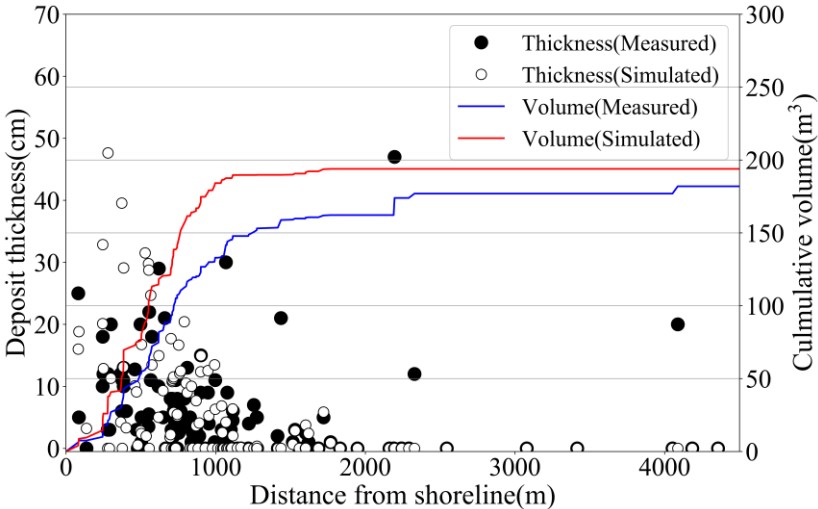

Figure 7 Comparison of field measured and simulated tsunami deposit thickness using a representa-tive grain size of $d = 0.127$ mm. Black point shows the measured thickness by Jankaew et al. (2008) and Gouramanis et al. (2017), white point shows the simulated thickness. Blue and red line show the cumulative curves of measured data and simulated data.

### 3.1.3.  *2004 IOT onshore sediment deposition*

In addition to the erosional features, the model simulated the deposition of 2004 IOT sediments across the island. The thickness of these simulated deposits is compared with 133 measured 2004 IOT deposit thicknesses (Jankaew et al., 2008; Gouramanis et al., 2017). Figure 7 shows a comparison of layer thicknesses at each site (black circles for measured results and white circles for simulated results), which shows that most of the sites are overestimated within 1 km from the shoreline and underestimated at distances greater than 2 km from the shoreline. The model specification and topographical data can be considered as the major causes of this error.

First, considering the overestimation within 1 km of the inundation distance, it is found that the STM has a setting of the maximum suspended concentration, $C_{max}$ as 37.7% (Xu, 1999a and 1999b). The computed suspended concentration in this area is higher than $C_{max}$. Therefore, the surplus sediment is forced to be deposited in this zone causing overestimation. Pham et al. (2018) found that the source of tsunami deposits in Phra Thong Island is mainly the sediment from the nearshore zone. In other words, the first wave, which had the highest wave height, eroded a large amount of sediment in the nearshore and transported a large amount of sediment inland. Therefore, it is considered that the maximum con-centration was reached during the first wave run-up because of the very high concentration of suspended sediment, which led to the overestimation of the forced sedimentation in the simulation.

Second, considering the underestimation of the deposition in inundation distances of 2 km or more, the most likely reason is the computational grid and the model specification. Previous studies have shown that tsunami deposits are highly affected by locality features (e.g. Sugawara et al., 2014a; Watanabe et al., 2018). As shown in three locations with the actual measured deposit thickness (dashed boxes) in Fig. 8, it can be seen that most of the measured thickness is zero which indicate and support

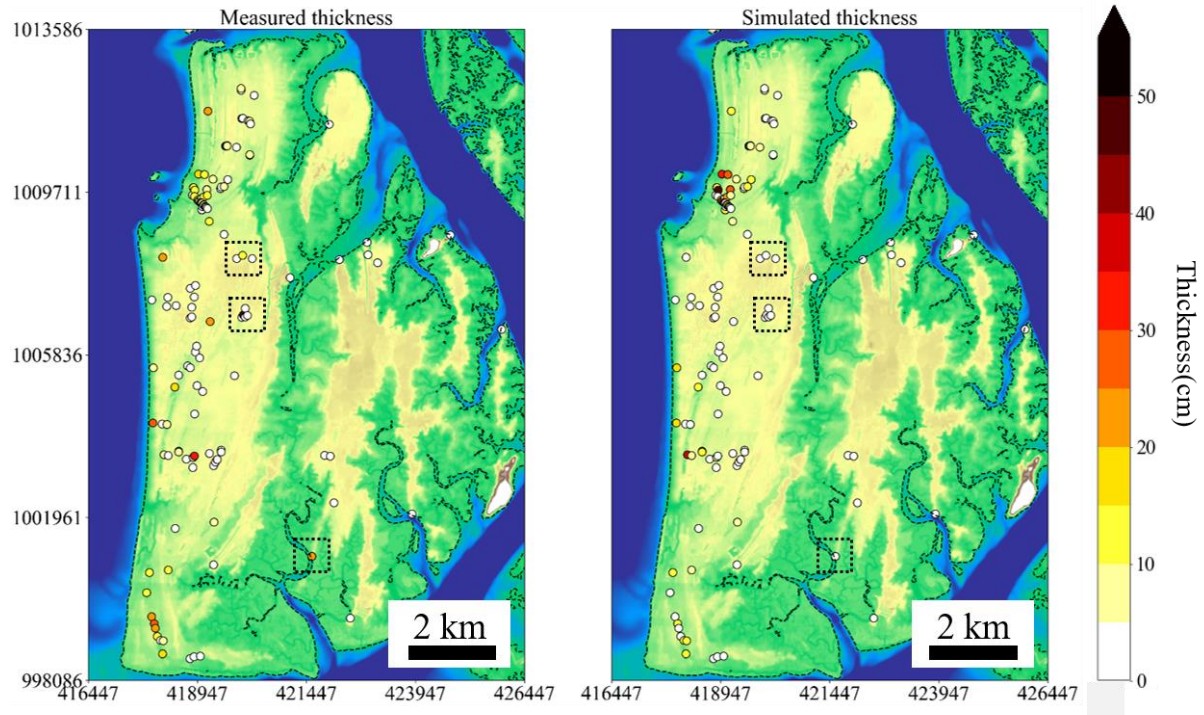

414

Figure 8 Spatial distribution of measured and simulated thickness of tsunami deposits. The black dotted

lines indicate that the calculated values are underestimated at distances greater than 2 km.

417

the reasons of localized deposition. Although the computational grid is very fine ($\Delta x_6$= 5 m), it is difficult to reproduce local sedimentation with averaged elevation data. It is worth noting that STM adopts only single grain size and can only perform deposits which consist sand. Sugawara et al. (2014a) conducted tsunami sediment transport simulation on the Sendai Plain and discussed the transportation possibility of finer grained sandy and muddy sediment. Muddy sediments were also found in the Sendai Plain at a distance of 2 km or more from the 2011 tsunami. Sugawara et al. (2014a)'s STM-based tsunami sediment transport simulation could not be reproduced for Phra Thong Island. This can be attribute to the limitation of sandy sediment and single grain size model. Therefore, it is possible that muddy or very fine-grained sediment was deposited at the three sites but were underestimated in the simulations using the current model.

Based on all above-mentioned reasons, it is more practical to evaluate the simulation results by the overall trend of the tsunami deposit rather than comparing the thickness point by point. In Figure 7, the line of "Cumulative volume" show the cumulative deposition expressed at each point by the sediment thickness multiplied by the area of the computational grid. In general, the tsunami deposits are greatly affected by local micro-topography (Sugawara et al., 2014a; Jaffe et al., 2016), and it is difficult to fit the modelled layer thickness with the observed layer thickness using DEM averaged in a computational grid. Therefore, we introduce the concept of cumulative sedimentation, and evaluated the scale of the amount of sediment movement generated. Although the modelled layer thickness typically overestimates the observed layer thickness by +7%, such low variation suggests a relatively successful

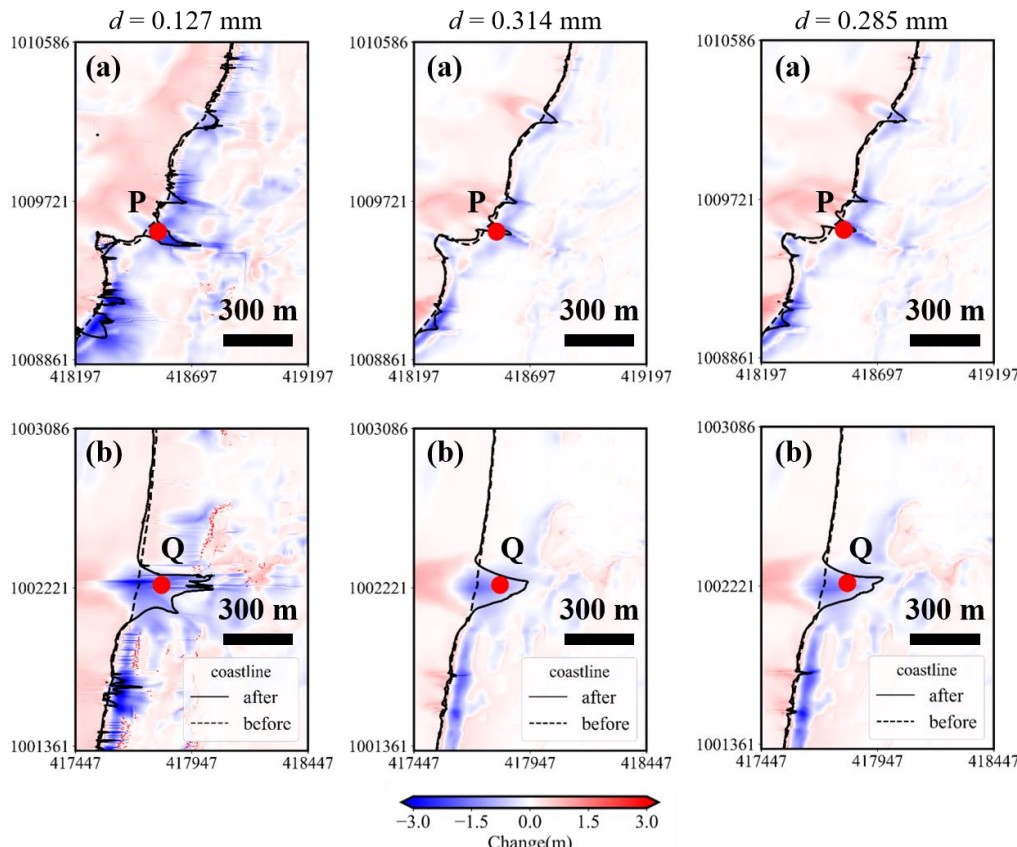


Figure 9 Topographic change and shoreline position caused by the tsunami for each grain size


Table 3 volume of erosion and deposition in regions (a) and (b) for each grain size (Percentage shows the ratio to reference)

|  | $d$ (mm) | Erosion (m³) | | Deposition (m³) | |
|---|---|---|---|---|---|
|  |  | Region (a) | Region (b) | Region (a) | Region (b) |
| Reference | 0.127 | 352,333 | 314,189 | 259,379 | 254,417 |
| Onshore | 0.285 | 143,793 (41%) | 155,225 (49%) | 161,810 (62%) | 149,470 (59%) |
| Offshore | 0.314 | 117,491 (33%) | 128,289 (41%) | 137,749 (53%) | 128,801 (51%) |


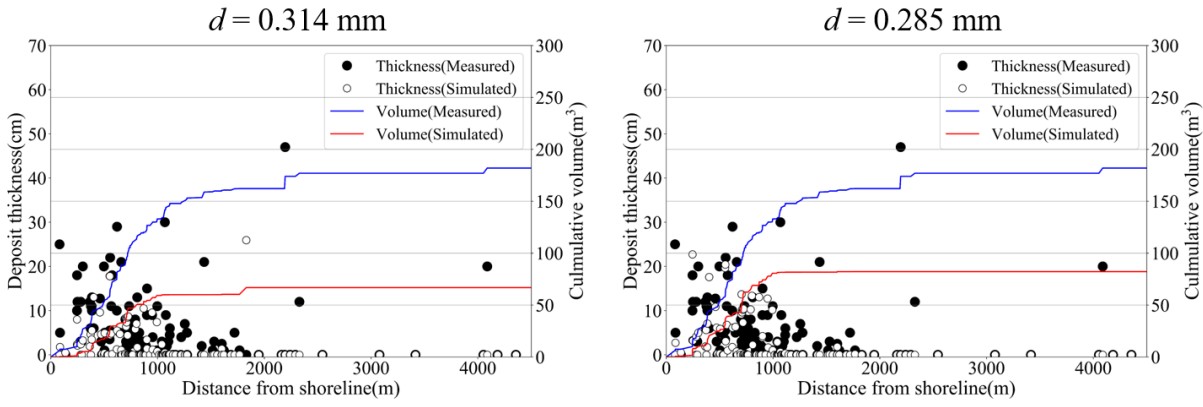


Figure 10 Comparison of field measured and simulated tsunami deposit thickness for each grain size.

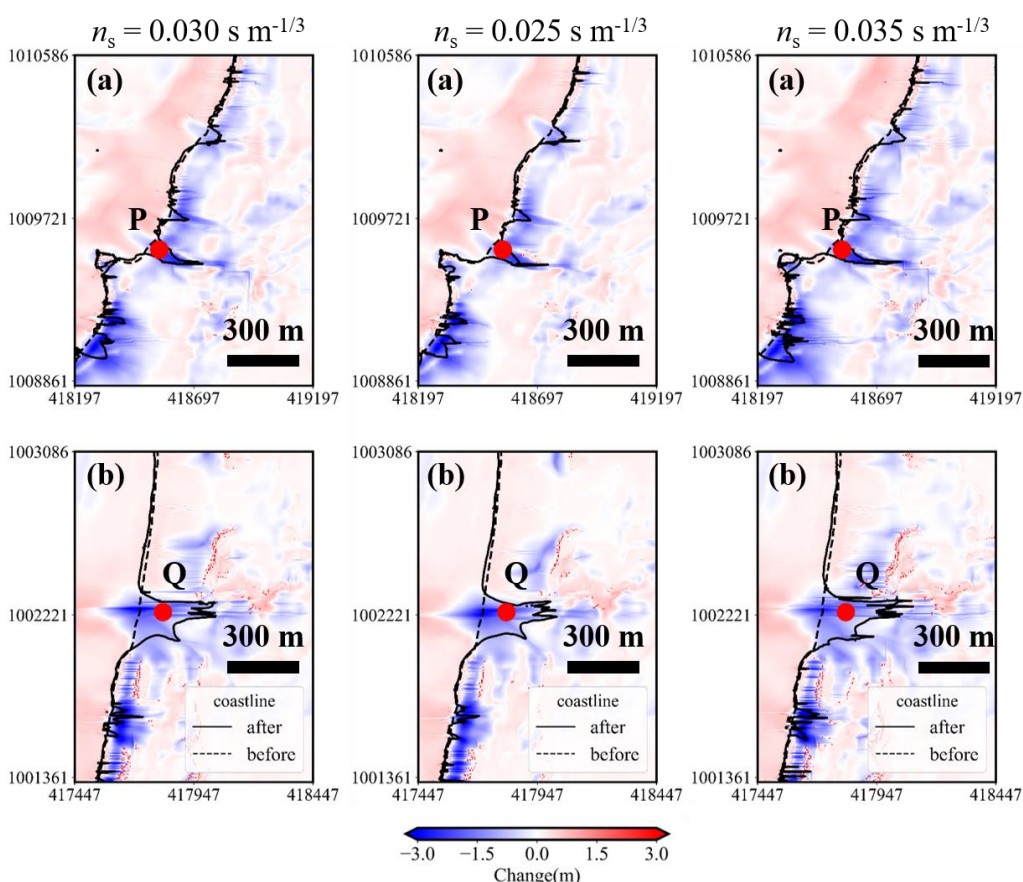

Figure 11 Topographic change and shoreline position caused by the tsunami for each roughness coefficient

Table 4 volume of erosion and deposition in regions (a) and (b) for each roughness coefficient (Percentage shows the ratio to reference)

| | $n_s$ (s m$^{-1/3}$) | Erosion (m$^3$) | | Deposition (m$^3$) | |
|---|---|---|---|---|---|
| | | Region (a) | Region (b) | Region (a) | Region (b) |
| Reference | 0.030 | 352,333 | 314,189 | 259,379 | 254,417 |
| Low | 0.025 | 293,032 (83%) | 285,659 (91%) | 249,242 (96%) | 230,905 (91%) |
| High | 0.035 | 410,323 (116%) | 352,284 (112%) | 272,394 (105%) | 262,522 (103%) |

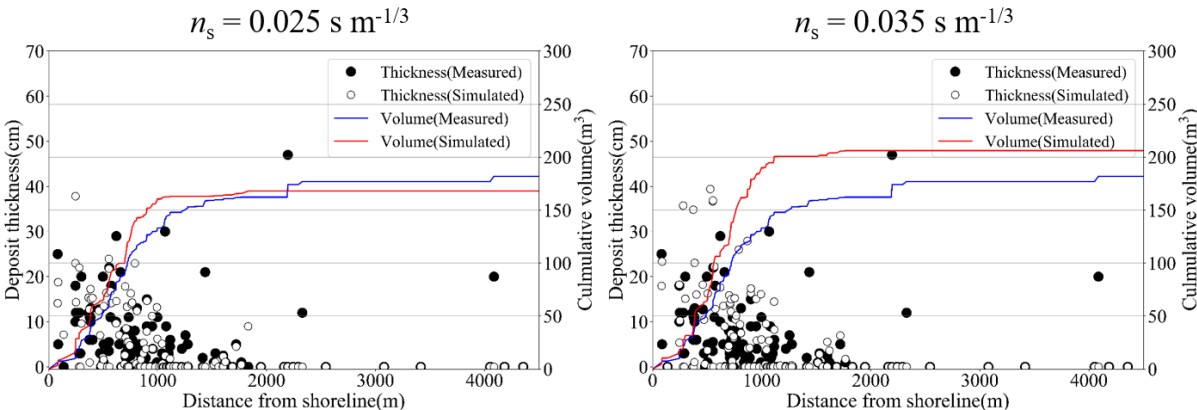

Figure 12 Comparison of field measured and simulated tsunami deposit thickness for each roughness coefficient.

reproduction of the observed dataset (Figure 7). The modelled overestimation is likely due to the assumption that the entire exposed land area would act as a movable bed. In reality, this is an oversimplification of the true ground surface, which contains vegetation that binds and traps the soil and wet regions (i.e. in swales) that would have higher degrees of sediment cohesion, reducing the area that would be eroded. In addition, the model also reproduces the inland thinning of the 2004 IOT deposit. Based on these results, comparison of the sediment layer thickness of the 2004 tsunami shows that the scale and the overall sediment transport trend are comparable, and therefore, the results are sufficiently reproducible with confidence to evaluate the actual sediment transport.

### 3.1.4. *Sensitivity analysis for grain size and roughness*

Figure 9 shows the topographical changes and the thickness of the sediment layers used in this calculation for each grain size, and Table 3 shows the volume of erosion and deposition in regions (a) and (b). These figures show that the smaller the grain size is, the greater the topographic change. This can be understood by the smaller the grain size, the larger the Shields parameter in Eq. (10), which indicates the ease of sediment transport, and the greater the amount of bed load in Eq. (8). However, Figure 9 suggests that the qualitative characteristics of sediment transport are the same in the three cases, due to the local erosion position of the beach in region (a) and (b) did not change for any grain size. And then, comparing the tsunami sediment thickness in Figure 10 the errors of the cumulative volume of $d = 0.314$ mm and $d = 0.285$ mm are -63% and -55%. Therefore, the grain size of $d = 0.127$ mm is considered to show the better reproducibility.

Figure 11 shows the topographical changes and thickness of sediment layer in this calculation for each bottom roughness coefficient, and Table 4 shows the volume of erosion and deposition in regions (a) and (b). These figures show that the larger the value of roughness coefficient $n_s$ is, the greater the topographic change. This can be understood by larger the roughness, the larger the Shields parameter in Eq. (10) because the friction velocity is proportionate to $n_s$. Therefore, an increase in the roughness coefficient indicates the ease of sediment transport, and the greater the amount of bed load in Eq. (8). However, Figure 11 suggests

that the qualitative characteristics of sediment transport are the same in the three cases, due to the local erosion position of the beach in region (a) and (b) did not change for any bottom conditions. And then, comparing the tsunami sediment thickness in Figure 12, the errors of the cumulative volume of $n_s$= 0.025 s m$^{-1/3}$ and $n_s$= 0.035 s m$^{-1/3}$ are -8% and 13%. Therefore, the roughness coefficient of $n_s$= 0.030 s m$^{-1/3}$ is considered to show the better reproducibility.

### 3.2. Sediment transport process

Although the model reproduces the zones of sediment erosion and deposition well, the sediment transport processes during the tsunami event are further examined in regions (a) and (b) in Figure 5. The modelled time series of the changes of water height and elevation at point P in region (a) and point Q in region (b) are shown in Figure 13. The modelling results show that the first wave arrived 2 hours 40 minutes after the earthquake, and backwash was generated 10 minutes later (Figure 13). In addition, the ground surface elevation increased by about 30 cm through sediment deposition during the first inflowing wave and more than 1.5 m was eroded during the backwash transporting sediment towards the ocean, so beach loss in both regions is considered to be a result of erosion during the backwash (red line in Figure 13).

In addition, no major topographic changes occurred on the beaches in both areas after the second wave backwashed, most of the sediment movement on the eroded beaches is considered to have been completed by the second wave drawback. In other words, the sediment transport processes during this period are the most important to examine the shoreline changes that occurred during the tsunami and set up the primary conditions for beach recovery post-tsunami. As such, there are two narrow time periods that highlight the key factors that for establishing the initial conditions of the recovery process. First, why was not the beach eroded by the inflowing waves? Second, how did the sediment flowing seaward in the first wave move?

Based on the waveform (which assumes a flat surface), a shore-normal cross section calculation was carried out along the transect in Figure 2. The transect covers the region (b) from 1,000 m offshore across the shoreline and 1,000 m inland. Beyond these distances the planar effect was considered to be negligible. Figure 14 shows the changes in ground level, water level, suspended sediment concentration and saturation of suspended sediment concentration on the transect at each unit of time as waves washed in and out.

### 3.2.1 Why was not the beach eroded by the pushing wave?

As shown in Figure 14, prior to the first wave, the ocean receded to below approximately 8 m below mean sea level. As inflow of the first wave began, sediment was eroded from the sea floor at about 5-10m below mean sea level. This nearshore erosion increased the suspended sediment concentration as the first wave propagated onshore. At the shoreline, the suspended sediment concentration saturated and sedimentation could begin at the shoreline. In other words, it is estimated that sediment eroded the nearshore (5 m < depth < 10 m) environment during the first inflowing wave, and much of this sediment

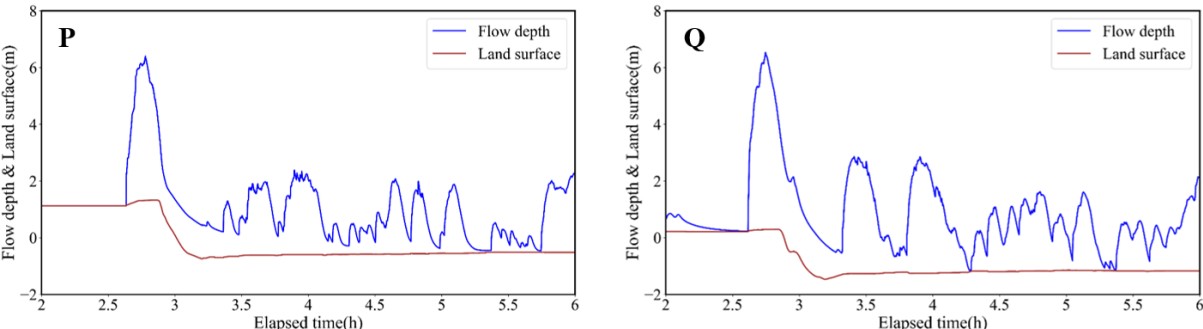


Figure 13 Chronological change of flow depth and land surface at point P and Q in region (a) and (b)

(blue line shows the flow depth and red line shows the land surface)


was transported shoreline and inland.

It should be noted that, there will be no increase in suspended sediment when the suspended sediment

is saturated in the model and is the likely reason that the beach was not eroded by the inflowing first
wave. Although there is a possibility that the beach was actually eroded, the numerical results suggest
that the erosion in shallow coastal waters (deeper than 5 m but shallower than 10 m) resulted in a very
high concentration of suspended sediment when the inflowing first wave entered -5 m to the beach
section of the coast and sediment ceased to be entrained. Pham et al. (2018) found that the source of the
2004 IOT deposits on Phra Thong Island was from the nearshore (depth < 15 m). This means that large
scale erosion in shallow water has occurred and a large amount of sediment has been transported inland
which agrees with the simulation results. Therefore, it is highly likely that the sediment concentration
was very high when it reached the beach during the first inflowing wave. Takahashi (2012) showed that
when the suspended sediment is in a high concentration state, turbulence is suppressed and the ability
to retain suspended sediment may decrease. Therefore, it is highly probable that the same phenomenon
occurred on Phra Thong Island and the beach erosion during the inflowing wave was suppressed.

**3.2.2 *How did the sediment flowing seaward in the first wave move?***

In Figure 14, at the initiation of backwash, the suspended sediment concentration is low. As backwash

flows towards the ocean, the velocity increases, which increases erosion and causes the suspended sed-
iment concentration to increase. This finding is consistent with the changes recorded in Figure 13. Beach
erosion due to backwash has also been confirmed in for the 2004 IOT in Sri Lanka and the 2011 Tsunami
along the Sendai Plain and at Rikuzentakata. (e.g. Tanaka et al., 2007, Tanaka et al. 2011, Yamashita et
al. 2015, 2016). On the Sendai Plain, the estuary section of the old river tends to increase the return
flow due to the tsunami (Tanaka et al., 2007, Tanaka et al. 2011). Therefore, there is a possibility that
the region (a) and (b) (Fig. 2, 5 and 6) where local beach erosion of the backwash occurred on Phra
Thong Island are the old river part.

Conversely, the entire beach was eroded by the return flow in Rikuzentakata (Yamashita et al., 2015,

2016), but no erosion was observed along the entire beach on Phra Thong Island and the Sendai Plain.
Yamashita et al. (2015, 2016) suggested that the difference between Rikuzentakata and the Sendai Plain

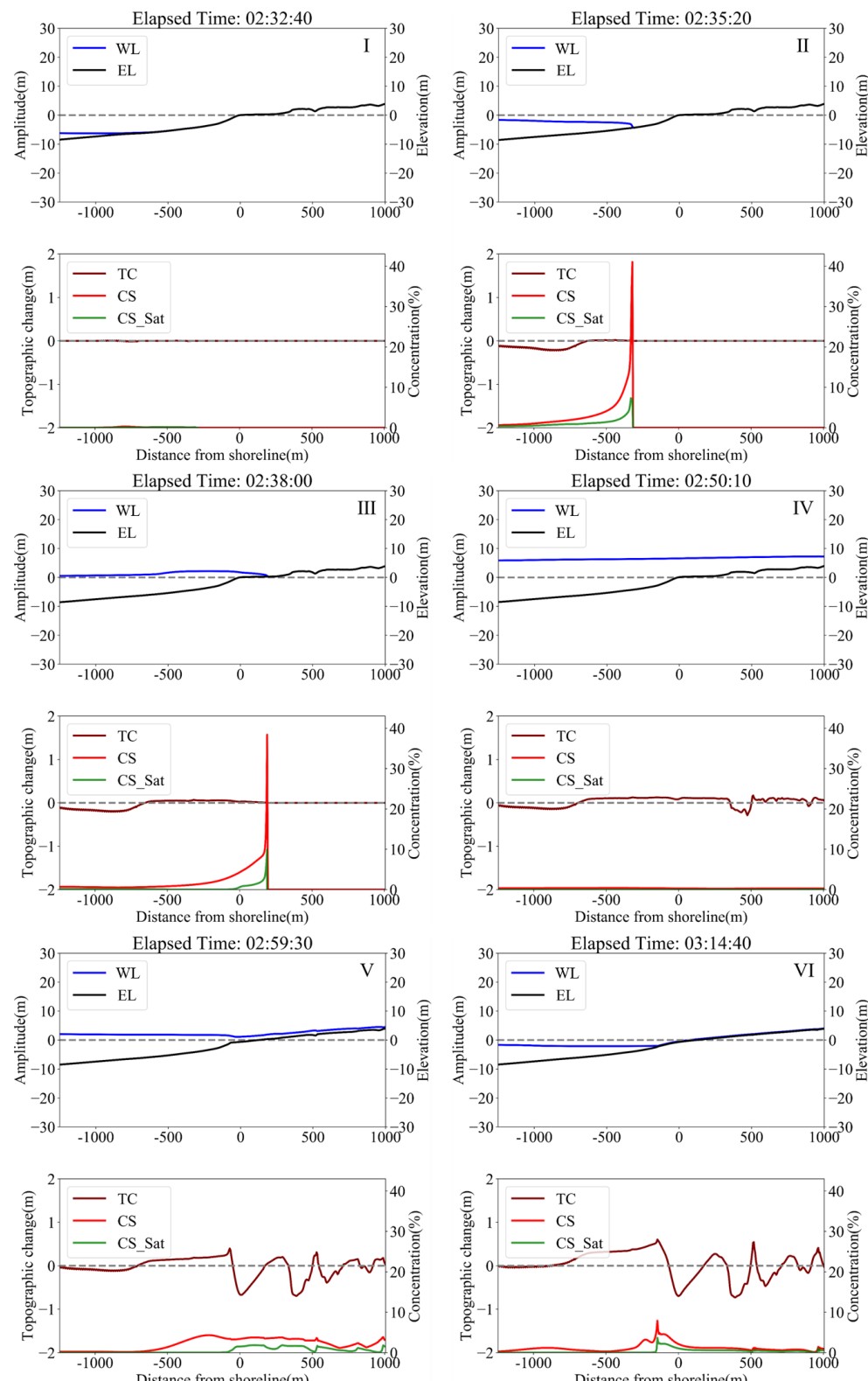

Figure 14 Change in water level (WL), land surface (EL), topographic change (TC), suspended sediment concentration (CS), and saturation suspended sediment concentration (CS_Sat) by section calculation along the survey line in region (b). ( I ) before the first inflowing wave, ( II ) Advance of first leading wave in shallow water, (III) Start of first leading wave run-up, (IV) Maximum of first leading wave , ( V ) Advance of second backwash, (VI) Maximum of second backwash

may be related to the horizontal distance of the plains. On the Sendai Plain, the inland topographic
gradient is small, the inundation distance is long and the inland inundation depth tends to be small.
Therefore, the potential energy that the inundation depth changes to kinetic energy during the backwash
(return flow) becomes relatively small. The Sendai Plain and Phra Thong Island are flooded plains over
2 km inland and have similar topographical features.
From the above reasons, the local beach erosion due to the return flow on Phra Thong Island occurred
at the mouths of tidal channels and within tidal channels and that minimal erosion occurred across the
wider beach ridge strand plain. As backwash of the first wave ended, the water still contained a high
suspended sediment concentration and this was deposited in the nearshore environment at less than 5
m water depth (Figure 15). After that, no significant topographic change was found. Thus, this model-
ling shows that most of the sediment that eroded from the onshore area was deposited in the shallow
nearshore zone.

**4.  Discussion**
**4.1.  *Sediment transport process and beach erosion***
Regions (a) and (b) were selected for detailed investigation of the simulation results and discussed.
On Phra Thong Island, the 2004 IOT wave was large enough to expose the nearshore sediments and
entrained most of its sediments from the shallow offshore region (below 5m). The wave ran up the
exposed nearshore area while retaining sediment from the shallow offshore region. The sediment con-
centration gradually increases as the wave runs up the relatively long distance of the exposed nearshore
zone, and became sediment-saturated as the wave reached the shoreline, making it difficult for new
sediment to be eroded further. This explains why there was little erosion of the beach during the inflow-
ing wave, and may be a characteristic sediment transport property of shallow beaches like those on Phra
Thong Island. The numerical simulation results suggest that there is little transportation of sediments
from beach by the first inflowing wave and that inland tsunami deposits originated from the nearshore
environment. This finding validates Sawai et al. (2009)'s observation that the 2004 IOT entrained dia-
toms from shallow offshore waters at Phra Thong Island, and Pham et al. (2018)'s observation that
sediment grain sizes and mineralogy were most similar to those of nearshore sediments. Figure 15
shows the results of the calculated sediment deposition both onshore and offshore Phra Thong Island.
From the modelling results, most of the eroded sediment was deposited in shallow nearshore environ-
ments in water less than approximately 5 m deep.
The simulations show that the eroded sediments were deposited in the nearshore zone during back-
wash (Fig. 15), which primed the coastal zone for rapid coastal recovery. The removal of sediment from
the onshore coastal zone also generated accommodation space that may have contributed to the coastal
recovery process. Future studies can build on these findings to determine the extent of sediment
transport and deposition, and identify the processes of coastal recovery on Phra Thong Island.
Geomorphologically, the Sendai Plain, which was inundated by the March 11, 2011 Great East Japan
tsunami, is similar to the beach ridge plain on Phra Thong Island (Tanaka et al., 2011), but most of the

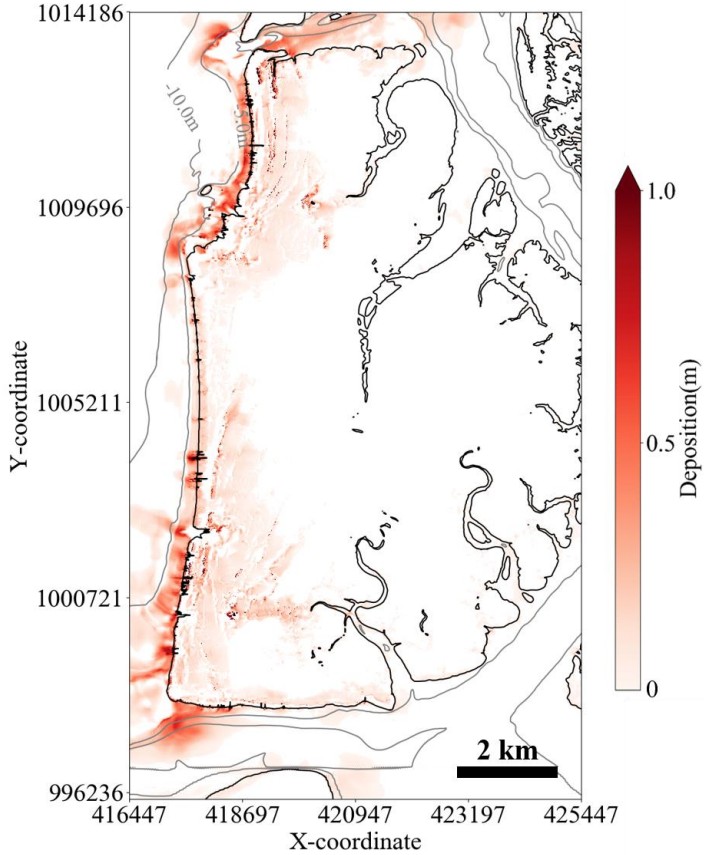

Figure 15 Sediment distribution derived from the simulation (showing depth contours at 5 m intervals
in the sea area)

tsunami sediment deposited onshore came from terrestrial sources (Goto et al., 2012; Szczuciński et al.,
2012; Takashimizu et al., 2012; Sugawara et al., 2014b). However, the Great East Japan tsunami differed
from the 2004 IOT as the Japanese event had a much smaller receding wave (Nationwide Ocean Wave
information network for Ports and HArbourS, NOWPHAS; available at
http://www.milt.go.jp/kowan/nowphas). As such the Japanese tsunami may not have achieved sediment
saturation as the wave approached the shoreline, thereby containing a lower sediment concentration and
allowing large volumes of sediment to be entrained from the beach for subsequent formation of inland
deposits. The different sources of deposited sediment in the two areas reflect contrasting sediment
transport mechanisms on shallow beaches, and may be useful for identifying paleotsunami from coastal
recovery and geological records.

**4.2.** *Limits of calculation results*
This study analyzed tsunami sediment transport on Phra Thong Island using numerical calculations
and assumed that the island was unvegetated and lacked topography. However, the western half of the
island has an undulating surface caused by the beach ridge and swale system, and is extensively vege-
tated with trees and dense grasses on the ridges and thick grasses within the swales. The eastern half of
the island has wide tidal channels and an extensive fringing mangrove system. Both topography and

differing vegetation types add complexity to the inundation and backflow sediment transport models not captured here. In future, it is necessary to consider the influence of vegetation and topography on tsunami sediment transport.

Another potential limitation of the model is the selection of a single (median) grain size for the sediments. As shown in previous studies (e.g. Sugawara et al., 2014a, b), the assumption of transport of single grain sized sediment differs from actual situations because of the distribution of grain sizes mobilized and deposited by tsunami. Therefore, it is important to set representative grain sizes and fully study how grain size affects tsunami sediment transport. Future modelling may consider simulating the suite of grain sizes individually or simulating a population of grain sizes that are identified in the modern environment and in preserved tsunami deposits.

Furthermore, although the calculation was performed considering the entire area a movable bed, the existence of fixed beds, such as rocky areas, should be considered. We consider this a minor component of this research as the rocky headlands that serve as fixed beds are relatively small in area and would contribute little to the overall simulations in our models.

Sugawara et al. (2014b) considers the simulation result of sediment layer thickness using the tsunami sediment transport calculation to be affected by grain size, bottom conditions and topographic data. Their study showed that the layer thickness increases as grain size becomes finer and the layer thickness distribution tendency was unchanged regardless of grain size. Similar results were obtained in this study.

## 5. Conclusion

Because of insufficient knowledge about the topographic recovery process after a tsunami, this study used sediment transport modelling to identify the erosional and depositional processes affecting the coastal zone at Phra Thong Island, Thailand during the 2004 Indian Ocean Tsunami.

First, it was confirmed by comparing simulated results of the shoreline and sediment layer thickness that the location of beach runoff identified on Phra Thong Island was reproducible and consistent with sediment transport results (Figs. 6 and 7). Based on the sediment transport results we conclude that the processes of sediment erosion and deposition on Phra Thong Island are characterized by the following sequence:

- erosion caused by the inflowing waves occurred at a relatively shallow location in the offshore area and the transported sediment was deposited near the shoreline;
- the inflowing waves caused minimal erosion of the shoreline; and,
- erosion of the shoreline was largely caused by backwash resulting in onshore sediments deposited in the shallow nearshore zone.

These erosional and depositional processes demonstrate the locations of sediment removal and subsequent deposition during the different phases of the first tsunami wave on Phra Thong Island which will serve as an important baseline of sediment sources for further study of the recovery process. The simulations also show that the zones of erosion and deposition across the island and offshore coastal

zone are non-uniform. In particular, the zones of erosion and deposition highlighted in the simulations
establish the environmental conditions that existed in the transitional phase between catastrophic tsu-
nami and normal coastal processes that facilitated coastal recovery.

## 6. Acknowledgements

We would like to express our gratitude for the support from Dr. Panon Latcharote of the Faculty of
Engineering, Mahidol University, Prof. Supot Teachavorasinskun, Dean of Faculty of Engineering,
Chulalongkorn University, Dr. Pitcha Jongvivatsakul, Department of Civil Engineering Chulalongkorn
University; and data from the Royal Thai Navy. RM, AS, KY, FI was support by JSPS Grant-in-Aid for
Scientific Research (A) No. 17H01631 (FY2017 - FY2021). AS and NL was support by JSPS Bilateral
program for joint research with National Research Council of Thailand (NRCT) (FY2017 - FY2018).
CG was supported by NUS grant (R-109-000-223-133). NL was supported by Ratchadapisek Sompoch
Endowment Fund (2019), Chulalongkorn University (762003-CC). This work is a contribution to IGCP
Project 639, 'Sea-level Change from Minutes to Millennia'.

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
