# Peer review of "Investigating beach erosion related with tsunami sediment transport at Phra Thong Island,"

_Natural Hazards and Earth System Sciences, 2019_

## Referee Comment (RC1) · Pedro Costa (Referee) · 30 Aug 2019

Dear authors, the manuscript by Masaya et al. presents a modeling exercise to better understand tsunami sedimentary dynamics on Phra Tong Island in Thailand.

The manuscript has quality and clearly has potential to be published after some moderate/major revisions. First of all, the title needs to be changed. It does not correspond to the work present and is confusing. In fact, there are occasional sentences that need to be rewritten (e.g. Lines 380 and following) and some spelling mistakes as well (e.g. palaotsunami instead of palaeotsunami). So, the manuscript clearly needs to be carefully revised by a native English speaker.

[Figure]

There are minor aspects that also need to be changed: Line 94 - change "conditions" to "setting" Line 254 to 256 - Move to Methods Line 292 - delet extra "." Figure 6 - Add scale and North arrow As I mentioned above, there are ocasional repetitions (e.g. "but that...but that...") that make the text less fluent. All these are minor aspects that should be corrected after detailed proof reading.

However, may main concern is related with the results (main source of tsunami sediment is the shallow offshore region) and the relevance of the Manning coefficient. These two aspects need to be fully revised and further discussed. It is common that tsunami deposits are feeded by dune and beach sediment (when I mention beach I am considering both the emerged and submerged areas). There are many examples that suggest tsunami deposits are essentially sourced once they started moving sediment in the shallower nearshore areas. The authors acknowledge that but they also mention the palaeontological content to support their reasoning. The fact is that paleontological content is from deeper regions but is transported and deposited by the tsunami because it is floating in the water column. So, you can have sediment from the coastal fringe and palaeontological from much deeper areas. These findinds are common and theydo not contradict each other. I suggest the authors make this clearer to the reader by adding a couple of sentences on this - clear definition of what offshore area is and clear definition of source. The other two main points I would like to raise are related with the use of an oversimplified grain-size coastal profile. You explain on the discussion the limitations of this approach but I strongly recommend that you make an attempt with varying grain-sizes according with the sedimentary environment - deep offshore; shallow offshore; beach (emerged and submerged); dune and depositional basin. What will be the changes if the grain-size varies in a way closer to reality - dune sediments are slightly finer than beach for example. What is the model response. I strongly advise the authors to do this because it would greatly improve the manuscript. I am aware a modeling exercise is a simplification of reality but it is important to not oversimplify things. Finally, I strongly recommend that you add a couple of sentences and present control tests on varying roughness coefficient. How does it affect the end

result?

Overall a nice manuscript that needs some slight improvments before it is accepted for publication. Therefore, and based on the above, I recommend this manuscript to be accepted for publication subject to major changes. Kind regards and congrats on a nice work Pedro JM Costa

---

## Referee Comment (RC2) · Anonymous Referee #2 · 30 Sep 2019

The comment was uploaded in the form of a supplement:
https://www.nat-hazards-earth-syst-sci-discuss.net/nhess-2019-263/nhess-2019-263-RC2-supplement.pdf

---

## Referee Comment (RC3) · Anonymous Referee #3 · 30 Sep 2019

The work "Investigating beach erosion related with its recovery at Phra Thong Island, Thailand caused by the 2004 Indian Ocean tsunami" addresses an interesting and challenging topic that is the scope of Natural Hazards Earth Systems Science. The work develops over a modelling approach to understand the short term coastal morphodynamics in relation to a tsunami event. Despite the challenging approach, the paper does not present consistent arguments that support the relation between the tsunami and beach erosion. In fact, in the abstract the authors conclude that "Our modelling approach confirms that beaches on Phra Thong Island were significantly eroded by the 2004 tsunami" but the analysis of the results, as displayed in figure 5, also show a lot of shoreline accretion. In fact, in most locations' shoreline seems to have experienced a

minor accretion (this is especially clear in figure 5a) while significant erosion is only observed at localized sections of the coast. In fact, the large longshore variability remains mostly unresolved, an should be further discussed in the manuscript. Although the hydrodynamic component of the work seems to portray a reasonable representation of the reality, the morphodynamic component is less robust and raises some questions.

1) The first statement of the conclusions "First, it was confirmed by comparing the measured and calculated values of the sediment layer thickness that the location of beach run off identified on Phra Thong Island was reproducible and consistent with sediment transport results", do not seem to have correspondence with the data presented in the paper.

2) In section 3.1.2 "change of shoreline" authors refer that sediment transport models confirm the erosion as portrayed by satellite images, but do not present satellite images before and after the tsunami occurrence. An objective comparison of model performance with satellite date with quantitative error statistics should also be present (e.g. brier skill score). The display of satellite images just before the tsunamic also would help the reader to have perception if the coastal embayments portrayed in image 6 existed before the tsunami.

3) The comparison of tsunami deposit thickness (figures 7 and 8) with the observed sediment layer also casts serious doubts on the model performance. In fact, the locations where the larger deposition were found (> 2000 inland) are the locations where the model predicted no accumulation. Moreover, a scatter plot with estimated layer thickness against observed thickness should be presented, supplemented with objective error statistics. Although authors discuss some discrepancies, this section should be expanded.

4) When comparing the model results with validation data, it seems that it would be more useful to present more detailed data, even though at a single site.

5) Concerning model application, there are lot simplifications that can affect model

results that are not properly justified or validated. Sediment transport magnitude and consequent morphological changes are largely dependent on the chosen values for the parameters displayed in table 4 . The assumption that some parameters assume a constant should also be justified namely the friction speed (or is this critical friction?) and bottom slope correction factor.

Minor comments:

a) line. 33 how can authors "confirm" if there is no observational data?

b) Line 73 – to support the statement "reproducibility has been confirmed by comparison between the calculated and measured values" a reference is needed.

c) figure 2 – a graphical scale or different gridline numbering should ease a better perception of the scale of the figure

d) line 234 – the use of " Manning's roughness coefficient was fixed at n = 0.025" contradicts the recognition (l438) that "bottom surface roughness greatly affects sediment transport"

e) lines228 to 239 – presents some formatting problems

f) Lines 238 – is the "limit Shields" is the critical Shields parameter? The authors should differentiate the Shields parameter from bottom shear stress (eq. 10)

g) Table 2 - The use of significant figures should be improved.

---

## Referee Comment (RC4) · Anonymous Referee #4 · 7 Oct 2019

This paper attempts to investigate erosion and deposition process as impacts of the 2004 Indian Ocean tsunami. Two sets of numerical modelling were used, namely TU-NAMI N2 and STM. The overall quality of the paper needs to be revised. Here are my major comments to the paper: 1. It is not clear how the TUNAMI N2 and STM were coupled. The authors need to provide more detailed information such as conformity of grid size, time step, and bathymetric-topography data. Furthermore, it is not clear whether the bed level in the TUNAMI N2 were also updated after sediment transport or not. 2. The reasons to run the simulation for 6 hours is not clear. Any data show the tsunami propagation at this area lasted in 6 hours? 3. The manning coefficient was treated uniform. Is the coefficient sensitive to the results? No specific sensitivity

analysis was done in this research. 4. This paper also attempts to bring recovery process of the beach, which I do not see where the recovery has taken place. Usually, beach recovery process takes years after a tsunami or storm surges. The impacts of the tsunami was performed by the models, but recovery process of the beach is not. Please see Section 4.1. How could the author relate these two processes? Lack of information of the method to observe the recovery and proofs of the recovery made this sub-topic not relevant to be discussed in this paper.

5. Backwash created deposition at the offshore area instead of erosion in other study area. But, this study revealed the opposite. Author needs to review some more cases that could give different result.

These are among literatures that proved differently:

Jiang, C., Chen, J., Yao, Y., Liu, J., and Deng, Y.: Study on threshold motion of sediment and bedload transport by tsunami waves, Ocean Eng., 100, 97–106, 2015.

Syamsidik, Al'ala, M., Fritz, H. M., Fahmi, M., and Hafli, T. M.: Numerical simulations of the 2004 Indian Ocean tsunami deposits' thicknesses and emplacements, Nat. Hazards Earth Syst. Sci., 19, 1265–1280, https://doi.org/10.5194/nhess-19-1265-2019, 2019.

Please discuss them when necessary.

My minor comments are as follows: 1. Diffusion coefficient in Equation 6 has different symbol in the paragraph explaining the equation; 2. Figure 10, three figures in the last row have no clear explanation: to what time these figures were meant to? Please provide sufficient information and discuss this properly.

---

## Author Comment (AC1) · 13 Feb 2020

**Dear Editor,**

Thank you for your time and sending us your decision. We have made corrections to both reviewers as shown below. Corrections made based on suggestions are shown in red.

**Reply to reviewer no. 1**
We highly appreciate the time spent for the review comments from the reviewer especially those minor corrections (our type errors) and pointed out many points that clarifications are needed. We are happy that the reviewer is happy and highly evaluated our manuscript. Please find our responses and corrections as shown below.

| Reviewer comments | Our answers | Corrected manuscript |
|---|---|---|
| -Title: The title needs to be changed.It does not corresponding to the work and is confusing. | Changed | Investigating beach erosion related with tsunami sediment transport at Phra Thong Island, Thailand caused by the 2004 Indian Ocean tsunami |
| - All the manuscript: occasional sentences that need to be rewritten(e.g. - Page 16 Line380 and following) and some spelling mistakes as well (e.g. palaotsunami instead of palaeotsunami) | Corrected | Please see the manuscript |
| - Page 3 Line 94: change "conditions" to "setting" | Corrected | Setting and methods |
| - Page 10 Line 254 - 256: Move to Methods | Moved(Page Line 254-256, Line 258-259, Line 265-266) to Methods | Please see the Methods(Page 7 Line 194– Page 8 Line 203) |
| - Page 11 Line 282:  delet extra "." | Corrected | |
| - Page 12 Figure 6: Add scale and North arrow | Added scale. Instead of Fig.6, we added it in Fig.1 and Fig.3. | Please see Page 13 Figure 6, Page 4 Figure 1, Page 6 Figure 3 |
| - I suggest the authors make this clearer to the reader by adding a couple of sentences on this - clear definition of what offshore area is and clear definition of source. | We defined offshore (water depth > 15 m) and nearshore (water depth < 15 m ) | Please see Page 10 Line 303 |
| - You explain on the discussion the limitations of this approach but I strongly recommend that you make an attempt with varying grain-sizes according with the sedimentary environment – deep offshore; shallow offshore; beach (emerged and submerged); dune and depositional basin. What will be the changes if the grain-size varies in a way closer to reality – dune sediments are slightly finer | We conducted the sensitive analysis of grain size. | Please see Page 10 Line 297-Line 308, Page 13 Line 372-379, Figs 8, 9,10 and Table 3. |

| | | |
|---|---|---|
| than beach for example. What is the model response. | | |
| - I strongly recommend that you add a couple of sentences and present control tests on varying roughness coefficient. How does it affect the end? | In general, when simulating tsunami sediment transport, it is necessary to determine the roughness coefficient according to land use.

However, since there is no land use map before the tsunami on Phra Thong Island, a fixed value was used, similar to previous studies (Yamashita et al., 2017; Yamashita et al., 2018).

Sugawara et al. (2014b) showed that the variation in Manning's roughness coefficient for the sand beds may affect the general distribution pattern of sediment deposits and erosions across the artificial topographic features.

Therefore, we do not analyze the sensitivity of Manning's roughness because Phra Thong Island has little artificial features. | Please see Page 10 Line277-285 |
| there are ocasional repetitions (e.g. "but that...but that...") that make the text less fluent. All these are minor aspects that should be corrected after detailed proof reading. | Corrected | Please see the manuscript |

---

## Author Comment (AC2) · 13 Feb 2020

**Dear Editor,**

Thank you for your time and sending us your decision. We have made corrections to both reviewers as shown below. Corrections made based on suggestions are shown in red.

**Reply to reviewer no. 2**

We highly appreciate the time spent for the review comments from the reviewer especially those minor corrections (our type errors) and pointed out many points that clarifications are needed. We are happy that the reviewer is happy and highly evaluated our manuscript. Please find our responses and corrections as shown below.

| Reviewer comments | Our answers | Corrected manuscript |
|---|---|---|
| - Page 2 Line 60-74: in terms of the sediment transport models induced by tsunami waves, the author should give certain credit to previous work (e.g. (Apotsos et al., 2011a; Apotsos et al., 2011b; Li et al., 2014) which use different sediment models while addressing similar problem. | Gave certain credit to previous work | Please see Page 3 Line 83-88 …(Takahashi et al., 1999; Gelfenbaun et al., 2007; Takahashi et al., 2008; Apotsos et al., 2011a; Apotsos et al., 2011b; Apotsos et al., 2011c; Takahashi et al., 2011; Gusman et al., 2012; Li et al., 2012; Takahashi et al., 2012; Li et al., 2014; Morishita & Takahashi, 2014; Yamashita et al., 2015; Yamashita et al., 2016; Arimitsu et al., 2017; Yamashita et al., 2017; Yamashita et al., 2018)… |
| - Page 2 Line 70: I'm not sure what "the movable bed model" refers to? Does it refer to a specific model or it represents all the sediment models assuming the bed is movable? If it refers to the former, then a definition is required to prepare the readers for the following context. | I made a mistake in the English translation. "Numerical modeling of tsunami sediment transport" is correct. | Please see Page 3 Line 82-83 …In recent years, the numerical modeling of tsunami sediment transport has been developed, … |
| - Page 3 Line 106-109: the presentation is confusing. Why using "Although…"? The second sentence seems contradictory with the first one. | Corrected. | Please see Page 3 Line 110-112 Due to the largely natural environment, Phra Thong Island is a rare case that is useful for verifying tsunami sediment transport models where few artificial features can generate model uncertainties. |
| Page 5-6, Section 2.3: about the tsunami source model, many source models have been proposed for the 2004 earthquake (e.g. (Banerjee et al., 2007; Chlieh et al., 2007; Grilli et al., 2007; Ioualalen et al., 2007; Rhie et al., 2007)). Different models could produce quite different tsunami wave heights in the same coastal area. Since the source model is | I wrote explaining why the current model is chosen. | Please see Page 5 Line 154-155 Suppasri et al.'s (2011) source model was focused on the coast of Thailand and accurately reproduced the inundation area and surveyed trace height of the 2004 IOT. |

| | | |
|---|---|---|
| one of the key factors which decide the reliability or accuracy of the simulation results, I feel the author should write a few sentences explaining why the current model is chosen. Does it produce better match with the measured data in this specific coast? | | |
| - Page 7-9 Section 2.4.2: about the "Tsunami movable bed model", two coefficients $\alpha$ and $\beta$ in formula (7) and (8) play significant role in the simulations, how these coefficients are specified? are the results sensitive to the choice of these coefficient? | Wrote the explaining detail. | Please see Page 9 Line 252 - 261

The grain-size dependent parameter for bed load ($\alpha$) and exchange rate ($\beta$) in Equation (9) and (10) are derived from Equations (12) and (13) based on the hydraulic experiments by Takahashi et al. (2011):

$$\alpha = 9.8044e^{-3.366d} \quad (12)$$

$$\beta = 0.0002e^{-6.5362d} \quad (13)$$

However, the functions should not be applied when $d$ is outside the 0.166 mm to 0.394 mm range as he validity of extrapolated $d$ values may produce erroneous results. |
| - Page 10 Section 3.1.1: How to define tsunami trace height? | "Tsunami height" is correct. Unified some expressions. | Please see Page 11 Section 3.1.1 |
| - Page Section 3, I feel the author tend to describe the result qualitatively instead of quantitatively, especially when mentioning the erosion and deposition results. Although the simulation results suffer from many uncertainties, I believe some quantitative explanation is necessary, e.g. the thickness of erosion or deposition Thickness | Added | Please see Page 14 Table 3 and Page 13 Line 354-356.

…Although the modelled layer thickness typically overestimates the observed layer thickness by +7%, such low variation suggests a relatively successful reproduction of the observed dataset (Figure 7)… |
| The figure quality needs to be improved, at least make sure the fontsize is consistent in all figures, not extra large (Figure 7-9) or extra small (Figure 10). Pay attention to the captain of each figure, make sure they are consistent with the legends inside the figure (see Figure 7 and Figure 8). | Revised | Please see all figures |

---

## Author Comment (AC3) · 13 Feb 2020

**Dear Editor,**

Thank you for your time and sending us your decision. We have made corrections to both reviewers as shown below. Corrections made based on suggestions are shown in red. 厳しめ

**Reply to reviewer no. 3**

We highly appreciate the time spent for the review comments from the reviewer especially those minor corrections (our type errors) and pointed out many points that clarifications are needed. We are happy that the reviewer is happy and highly evaluated our manuscript. Please find our responses and corrections as shown below.

| Reviewer comments | Our answers | Corrected manuscript |
|---|---|---|
| - All the manuscript: In fact, in the abstract the authors conclude that "Our modelling approach confirms that beaches on Phra Thong Island were significantly eroded by the 2004 tsunami" but the analysis of the results, as displayed in figure 5, also show a lot of shoreline accretion. In fact, in most locations' shoreline seems to have experienced a minor accretion (this is especially clear in figure 5a) while significant erosion is only observed at localized sections of the coast. In fact, the large longshore variability remains mostly unresolved, an should be further discussed in the manuscript. | Our expression was bad. We focused on beach erosion in two areas(region (a) and (b)) that have been locally eroded, but not all beaches in Phra Thong Island. | Please see Page 1 Line33-34 …Our modelling approach suggests that beaches located in two regions on Phra Thong Island were significantly eroded by the 2004 tsunami…

 Please Page 5 Figure 2 Caption: Figure 2 Terrain data (The black frame shows Region 1 to Region 6, and the black line in Region 6 shows the cross-section where calculation was performed. Dashed squares are the beach where erosion was confirmed from satellite image.) |
| The first statement of the conclusions "First, it was confirmed by comparing the measured and calculated values of the sediment layer thickness that the location of beach run off identified on Phra Thong Island was reproducible and consistent with sediment transport results", do not seem to have correspondence with the data presented in the paper. | Corrected | Please see Page 21 Line 507-509

 First, it was confirmed by comparing simulated results of the shoreline and sediment layer thickness that the location of beach runoff identified on Phra Thong Island was reproducible and consistent with sediment transport results. |
| In section 3.1.2 "change of shoreline" authors refer that sediment transport models confirm the erosion as portrayed by satellite images, but do not present satellite images before and after the tsunami occurrence. An objective comparison of model performance with satellite date with quantitative error | Thank you for the suggestion. We tried to use images prior the tsunami, but limited resolution of our images caused the difficulty for determing the shoreline.
 Therefore, shoreline from bathymetry data was added to provide lack information of images. | Please see Page 13 Figure 6 Caption: …satellite image (30 Jan, 2005), which is overlain by the modelled extent of erosion showing that the modelled results closely match the observed changes. The red line is the calculated shoreline after the tsunami, and the blue line is the shoreline before the tsunami… |

| | | |
|---|---|---|
| statistics should also be present (e.g. brier skill score). The display of satellite images just before the tsunamic also would help the reader to have perception if the coastal embayments portrayed in image 6 existed before the tsunami. | | |
| The comparison of tsunami deposit thickness (figures 7 and 8) with the observed sediment layer also casts serious doubts on the model performance. In fact, the locations where the larger deposition were found (> 2000 inland) are the locations where the model predicted no accumulation. Moreover, a scatter plot with estimated layer thickness against observed thickness should be presented, supplemented with objective error statistics. Although authors discuss some discrepancies, this section should be expanded. | We introduced the concept of cumulative sedimentation, and evaluated the scale of the amount of sediment movement generated. | Please see Page 13 Line 350-371 and Figure 7

The line of "volume" show the cumulative deposition expressed at each point by the sediment thickness multiplied by the area of the computational grid. In general, the tsunami deposits are greatly affected by local micro-topography(Sugawara et al., 2014; Jaffe et al., 2016), and it is difficult to fit the modelled layer thickness with the observed layer thickness using DEM averaged in a computational grid. Therefore, we introduced the concept of cumulative sedimentation, and evaluated the scale of the amount of sediment movement generated. Although the modelled layer thickness typically overestimates the observed layer thickness by +7%, such low variation suggests a relatively successful reproduction of the observed dataset (Figure 7). |
| When comparing the model results with validation data, it seems that it would be more useful to present more detailed data, even though at a single site. | Thank you for your comment. | |
| Concerning model application, there are lot simplifications that can affect model results that are not properly justified or validated. Sediment transport magnitude and consequent morphological changes are largely dependent on the chosen values for the parameters displayed in table 4 . The assumption that some parameters | In tsunami sediment transport model, uncertainty of those parameters were often simplified for simulation.
 Based on previous studies, those parameters were generally given by fixed value which were also used in this study.
 Parameters were justified namely critical friction. | Please see Page 10 Section 2.5. and Table 2. |

| | | |
|---|---|---|
| assume a constant should also be justified namely the friction speed (or is this critical friction?) and bottom slope correction factor. | | |
| - Page1 Line 33: how can authors "confirm" if there is no observational data? | Changed "suggests" | Our modelling approach  suggests that beaches … |
| - Page 1 Line 73: to support the statement "reproducibility has been confirmed by comparison between the calculated and measured values" a reference is needed. | Corrected | Please Page 3 Line 88-89

Yamashita et al., 2015; Yamashita et al., 2016; Arimitsu et al., 2017; Yamashita et al., 2017; Yamashita et al., 2018 |
| - Page 5 Figure 2: a graphical scale or different gridline numbering should ease a better perception of the scale of the figure. | Added | Please Page 5 Figure 2 |
| - Page 9 Line 234: the use of " Manning's roughness coefficient was fixed at n = 0.025" contradicts the recognition (l438) that "bottom surface roughness greatly affects sediment transport" | Thank you for your comment on related issue. Fixed value of coefficient were used in this study because of no land use map were available in this area. | Please see Page 10 Line 277-285 |
| - Page 9 Line 228-239: presents some formatting problems | Corrected | Please see Page 10 Line 270-289 |
| - Page 9 Lines 238: is the "limit Shields" is the critical Shields parameter? The authors should differentiate the Shields parameter from bottom shear stress (eq. 10) | Corrected | The  Critical Shields number … |
| - Page 10 Table 2 - The use of significant figures should be improved | Corrected | See Page 11 Table 2 |

---

## Author Comment (AC4) · 13 Feb 2020

**Dear Editor,**

Thank you for your time and sending us your decision. We have made corrections to both reviewers as shown below. Corrections made based on suggestions are shown in red. かなり丁寧に査読コメントをくれた印象.

**Reply to reviewer no.4**

We highly appreciate the time spent for the review comments from the reviewer especially those minor corrections (our type errors) and pointed out many points that clarifications are needed. We are happy that the reviewer is happy and highly evaluated our manuscript. Please find our responses and corrections as shown below.

| Reviewer comments | Our answers | Corrected manuscript |
|---|---|---|
| - It is not clear how the TUNAMI N2 and STM were coupled. The authors need to provide more detailed information such as conformity of grid size, time step, and bathymetric-topography data. Furthermore, it is not clear whether the bed level in the TUNAMI N2 were also updated after sediment transport or not. | I wrote the explaining. | Please Page 8 Line 209-211

For each time step, the STM receives the total flow fluxes from TUNAMI-N2 and calculates the change of seafloor and land surface and feeds this to the next time step of the TUNAMI-N2 model. |
| - The reasons to run the simulation for 6 hours is not clear. Any data show the tsunami propagation at this area lasted in 6 hours?
なぜ再現時間 6 時間で計算したのか？ | Added the explaining. | Please see Page 10 Line274-276

The simulations were calculated over a 0.05 second increment with a 6 hour period in which the test case with a 12 hour period showed the suspended sediment concentration in the vicinity of the shoreline decreased and stabilized. |
| - The manning coefficient was treated uniform. Is the coefficient sensitive to the results? No specific sensitivity analysis was done in this research. | Fixed value of coefficient were used in this study sue to land use map are not available in this area.
The lack understanding of manning roughness coefficient will be mainly discussed as issue by creating city use maps through field surveys in the future. | Please see Page 10 277-285 |
| This paper also attempts to bring recovery process of the beach, which I do not see where the recovery has taken place. Usually, beach recovery process takes years after a tsunami or storm surges. The impacts of the tsunami was performed by the models, but recovery process of the beach is not.. | The terrain recovery after tsunami is determined by the factors of coastal conditions which used as initial conditions for tsunami movement simulation.
In this study, we aimed to clarify the types of sediment movement which was caused by tsunami and the correlated initial condition.
This study highlighted that out flowing sand was relatively easy to return to shoreline. | |

| | Whether it has actually returned will be examined in the future using wave / wind data / wave models.  And recovery process of the beach will be mainly discussed as an issue in future study. | |
| --- | --- | --- |
| - Backwash created deposition at the offshore area instead of erosion in other study area. But, this study revealed the opposite. Author needs to review some more cases that could give different result. | Thank you for your advice. In the future, we plan to study other areas. | |
| - Diffusion coefficient in Equation 6 has different symbol in the paragraph explaining the equation | Corrected | Tsunami  height   Tsunami height |
| - Figure 10, three figures in the last row have no clear explanation: to what time these figures were meant to? Please provide sufficient information and discuss this properly. | Added | Please Page 18 Figure 12 |

---

## Referee Report (RR1)

[referee-annotated manuscript omitted]

---

## Author Response (AR2)

**Dear Editor,**

Thank you for your time and sending us your decision. We have made corrections to both reviewers as shown below. Corrections made based on suggestions are shown in red.

**Reply to reviewer no. 1**

We highly appreciate the time spent for the review comments from the reviewer especially those minor corrections (our type errors) and pointed out many points that clarifications are needed. We are happy that the reviewer is happy and highly evaluated our manuscript. Please find our responses and corrections as shown below.

| Reviewer comments | Our answers | Corrected manuscript |
|---|---|---|
| -Title: The title needs to be changed.It does not corresponding to the work and is confusing. | Changed | Please see title

Investigating beach erosion related with tsunami sediment transport at Phra Thong Island, Thailand caused by the 2004 Indian Ocean tsunami |
| - All the manuscript: occasional sentences that need to be rewritten(e.g. - Page 16 Line380 and following) and some spelling mistakes as well (e.g. palaotsunami instead of palaeotsunami) | Corrected | The manuscript has been fully checked again including grammar and spell check. |
| - Page 3 Line 94: change "conditions" to "setting" | Corrected | Setting and methods (Page 4 Line 116) |
| - Page 10 Line 254 - 256: Move to Methods | Moved(Page Line 254-256, Line 258-259, Line 265-266) to Methods | Please see the 2.4.1 (Page 7 Line 192– Page 8 Line 201) |
| - Page 11 Line 282: delet extra "." | Corrected | |
| - Page 12 Figure 6: Add scale and North arrow | Added scale. Instead of Fig.6, we added it in Fig.1 and Fig.3. | Please see Figure 1, Figure 3 |
| - I suggest the authors make this clearer to the reader by adding a couple of sentences on this - clear definition of what offshore area is and clear definition of source. | We defined offshore (water depth > 15 m) and nearshore (water depth < 15 m ) | Please see Page 12 Line 335

… the offshore (water depth > 15 m), nearshore (water depth < 15 m ), and onshore … |
| - You explain on the discussion the limitations of this approach but I strongly recommend that you make an attempt with varying grain-sizes according with the sedimentary environment – deep offshore; shallow offshore; beach (emerged and submerged); dune and depositional basin. What will be the changes if the grain-size varies in a way closer to reality – | Based on Pham et al.(2018), we conducted the sensitivity analysis of grain size by performing the simulation using three grain sizes, 0.127 mm, 0.314 mm and 0.285 mm.

From the sensitivity analysis result, it can be seen that smaller grain size causes larger bathymetric change (Table 3) but locations of erosions and | Please see Figs 9,10 and Table 3 and additional explanations from the sensitivity analysis below.

Page 11 Line 326-Line 340,
The numerical model used in this paper can only consider a single grain size, so the model cannot resolve the grading of the sand layer. Additionally, initial bed grain size can have a large effect |

| | | |
|---|---|---|
| dune sediments are slightly finer than beach for example. What is the model response. | bathymetric changes in both regions (a) and (b) still have the same trend. In addition, error of the cumulative volume for d = 0.127 mm is the smallest (+7 %) while -55 % and -63 % for d = 0.285 mm and d = 0.314 mm respectively. Therefore, we judged that using one single grain size (d = 0.127 mm) is reliable and suitable for further discussion in later of the paper. | on erosion and deposition (e.g. Apotsos et al., 2011; Sugawara et al., 2014a; Jaffe et al., 2016). Furthermore, the sediment data we used to set the grain size is only one point on the north side, so it cannot be said that it is sufficient data to set the representative grain size. Therefore, it is necessary to perform sensitivity analysis on grain size.

Page 18 Line 453-463 (section 3.1.4)
Figure 9 shows the topographical changes and thickness of sediment layer in this calculation for each grain size, and Table 3 shows the volume of erosion and deposition in regions (a) and (b). These evidences show that smaller the particle size is, the greater the topographic change. This can be understood by the smaller the particle size, the larger the Shields number in Eq. (10), which indicates the ease of sediment transport, and the greater the amount of bed load in Eq. (8). However, Figure 9 suggested that the qualitative characteristics of sediment transport are same in the three cases, due to the local erosion position of the beach in region (a) and (b) did not change for any particle size. And then, comparing the tsunami sediment thickness in Figure 10 the errors of the cumulative volume of d = 0.314 mm and d = 0.285 mm are -63% and -55%. Therefore, the grain size of d = 0.127 mm is considered to show the better reproducibility. |
| - I strongly recommend that you add a couple of sentences and present control tests on varying roughness coefficient. How does it affect the end? | We have also tested sensitivity analysis for additional two bottom coefficients at $n_s$ = 0.025 s/m$^{1/3}$ and $n_s$ = 0.035 s/m$^{1/3}$ which are within the range of commonly used in previous studies(e.g. Sugawara et al., 2014a, b)

From the sensitivity analysis result, it can be seen that smaller grain size causes larger | Please see Figs 11,12 and Table 4 and additional explanations from the sensitivity analysis below.

Page 11 Line 302-314
For the bottom conditions of STM, the roughness coefficient was fixed at $n_s$ = 0.030 s/m$^{1/3}$, and the entire area of Region 6 was considered the movable bed. In general, when simulating tsunami |

bathymetric change (Table 4) but locations of erosions and bathymetric changes in both regions (a) and (b) still have the same trend. In addition, error of the cumulative volume for ns = 0.030 is the smallest (+7 %) while -8 % and +13 % for $n_s$ = 0.025 and $n_s$ = 0.035 respectively. Therefore, we judged that using one single roughness coefficient (ns = 0.030) is reliable and suitable for further discussion in later of the paper.

sediment transport, it is necessary to determine the roughness coefficient according to land use. However, since there is no land use map before the tsunami on Phra Thong Island, a fixed value was used, similar to previous studies (e.g. Sugawara et al., 2014a, b; Yamashita et al., 2015; Yamashita et al., 2016). However, Sugawara et al. (2014a) showed that the variation in Manning's roughness coefficient for the sand beds may affect the general distribution pattern of sediment deposits and erosions across the artificial topographic features with much higher roughness coefficient such as artificial canals, roads and populated residential areas. Therefore, a sensitivity analysis on the roughness coefficient was performed. Phra Thong Island has no such artificial topographic features and using the single roughness coefficient should sufficiently capture the overall roughness. However, to ensure robust conclusions, a sensitivity analysis for two bottom conditions was performed at $n_s$ = 0.025 s/m$^{1/3}$ and $n_s$ = 0.035 s/m$^{1/3}$, which are within the range of previously used estimates of roughness (e.g. Sugawara et al., 2014a, b).

Page 18 Line 464-474,
Figure 11 shows the topographical changes and thickness of sediment layer in this calculation for each bottom roughness coefficient, and Table 4 shows the volume of erosion and deposition in regions (a) and (b). These evidences show that larger the value of roughness coefficient n_s is, the greater the topographic change. This can be understood by larger the roughness, the larger the Shields number in Eq. (10) because the friction velocity is proportionate to ns. Therefore, an increase in the roughness coefficient indicates the ease of sediment transport, and the greater the amount of bed load in

| | | Eq. (8). However, Figure 11 suggested that the qualitative characteristics of sediment transport are same in the three cases, due to the local erosion position of the beach in region (a) and (b) did not change for any bottom conditions. And then, comparing the tsunami sediment thickness in Figure 12, the errors of the cumulative volume of ns= 0.025 s/m1/3 and ns= 0.035 s/m1/3 are -8% and 13%. Therefore, the roughness coefficient of ns= 0.030 s/m1/3 is considered to show the better reproducibility. |
|---|---|---|
| there are ocasional repetitions (e.g. "but that...but that...") that make the text less fluent. All these are minor aspects that should be corrected after detailed proof reading. | Corrected | We have also checked and fixed this issue. Please see the manuscript |

**Dear Editor,**

Thank you for your time and sending us your decision. We have made corrections to both reviewers as shown below. Corrections made based on suggestions are shown in red.

**Reply to reviewer no. 2**
We highly appreciate the time spent for the review comments from the reviewer especially those minor corrections (our type errors) and pointed out many points that clarifications are needed. We are happy that the reviewer is happy and highly evaluated our manuscript. Please find our responses and corrections as shown below.

| Reviewer comments | Our answers | Corrected manuscript |
|---|---|---|
| - Page 2 Line 60-74: in terms of the sediment transport models induced by tsunami waves, the author should give certain credit to previous work (e.g. (Apotsos et al., 2011a; Apotsos et al., 2011b; Li et al., 2014) which use different sediment models while addressing similar problem. | As advised, we addressed more credit to previous works.

STM
Takahashi et al., 2000; Takahashi et al., 2008; Takahashi et al., 2011; Gusman et al., 2012; Ranasinghe et al., 2013; Morishita and Takahashi, 2014; Yamashita et al., 2015; Yamashita et al., 2016; Arimitsu et al., 2017; Yamashita et al., 2017; Yamashita et al., 2018)
XBeach
(Li et al., 2012; Li et al., 2013; Li et al., 2014)
Delft3D
(Gelfenbaun et al., 2007; Apotsos et al., 2011a; Apotsos et al., 2011b; Apotsos et al., 2011c). | Please see Page 3 Line 79-85.

…has been developed (e.g. Takahashi et al., 2000), improved (e.g. Takahashi et al., 2011; Apotsos et al., 2011a; Li et al., 2013; Morishita and Takahashi, 2014; Yamashita et al., 2018) and applied in the field (e.g. Gelfenbaun et al., 2007; Takahashi et al., 2008; Apotsos et al., 2011b; Apotsos et al., 2011c; Gusman et al., 2012; Li et al., 2014; Arimitsu et al., 2017; Yamashita et al., 2017), and reproducibility has been confirmed by comparison between the calculated and measured values (e.g. Li et al., 2012; Ranasinghe et al., 2013; Sugawara et al., 2014a; Yamashita et al., 2015; Yamashita et al., 2016). |
| - Page 2 Line 70: I'm not sure what "the movable bed model" refers to? Does it refer to a specific model or it represents all the sediment models assuming the bed is movable? If it refers to the former, then a definition is required to prepare the readers for the following context. | We are sorry for a mistake in the English translation. "Numerical modeling of tsunami sediment transport" is correct. | Please see Page 3 Line 78-79

…In recent years, the numerical modeling of tsunami sediment transport has been developed, … |
| - Page 3 Line 106-109: the presentation is confusing. Why using "Although…"? The second sentence seems contradictory with the first one. | Corrected. | Please see Page 3 Line 100

Due to the largely natural environment, Phra Thong Island is a rare case that is useful for verifying tsunami sediment transport models where few artificial features can generate model uncertainties. |
| Page 5-6, Section 2.3: about the tsunami source model, many | We added an explanation why the current model is chosen. | Please see Page 6 Line 159-160 |

| | | |
|---|---|---|
| source models have been proposed for the 2004 earthquake (e.g. (Banerjee et al., 2007; Chlieh et al., 2007; Grilli et al., 2007; Ioualalen et al., 2007; Rhie et al., 2007)). Different models could produce quite different tsunami wave heights in the same coastal area. Since the source model is one of the key factors which decide the reliability or accuracy of the simulation results, I feel the author should write a few sentences explaining why the current model is chosen. Does it produce better match with the measured data in this specific coast? | The tsunami source model proposed by Suppasri et al. (2011) was selected because this source model was originally proposed using only detail observation (waveforms) and survey (runup heights) data in Thailand for reproducing the 2004 tsunami in Thailand. Therefore, we believe that this source model is the most suitable for applying to our study area in Thailand. | …as the model focused on the coast of Thailand and accurately reproduced the inundation area and surveyed trace height of the 2004 IOT. |
| - Page 7-9 Section 2.4.2: about the "Tsunami movable bed model", two coefficients $\alpha$ and $\beta$ in formula (7) and (8) play significant role in the simulations, how these coefficients are specified? are the results sensitive to the choice of these coefficient? | We have added the explanations detail.
These parameters were determined the hydraulic experiments by Takahashi et al. (2011) using three grain size, d = 0.166 mm, 0.267 mm and 0.394 mm. Equations (11) and (12) are used to interpolate $\alpha$ and $\beta$ when the grain size is not exactly the same as the experiment. We assumed that our grain size (0.127 mm) is applicable as it is slightly smaller than the smallest range of the experiment.
Selection of these parameters may affect to the simulation results similar to the change of grain size. For this, we have tested sensitivity analysis (section 3.1.4) for different grains size and roughness coefficients. | Please see Page 9 Line 248 – 256

The grain-size dependent parameter for bed load ($\alpha$) and exchange rate ($\beta$) in Equation (8) and (9) are derived from Equations (11) and (12) based on the hydraulic experiments by Takahashi et al. (2011):

$$\alpha = 9.8044e^{-3.366d} \quad (11)$$

$$\beta = 0.0002e^{-6.5362d} \quad (12)$$

However, the functions should not be applied when $d$ is outside the 0.166 mm to 0.394 mm range as he validity of extrapolated $d$ values may produce erroneous results. |
| - Page 10 Section 3.1.1: How to define tsunami trace height? | It should be the maximum tsunami height which is measured as the maximum water elevation from the mean sea-level. | Please see Page 12 Line 346-347

… of the maximum tsunami heights and the seven measured tsunami heights on … |
| - Page Section 3, I feel the author tend to describe the result qualitatively instead of quantitatively, especially when mentioning the erosion and | There are two quantitative discussions in the revised manuscript 1) Cumulative volume and 2) Sensitivity analysis | Page 16 Line 425-432
the line of "Cumulative volume" show the cumulative deposition expressed at each point by the sediment thickness multiplied by |

| | | |
|---|---|---|
| deposition results. Although the simulation results suffer from many uncertainties, I believe some quantitative explanation is necessary, e.g. the thickness of erosion or deposition Thickness | 1) Cumulative volume is the total volume (summation) of sediment thickness at all surveyed and simulated points. This can evaluate the size or total amount of the transported sediment. (section 3.1.3)

2) Sensitivity analysis by varying grain sizes and roughness coefficients have been made to evaluate volume of erosion and deposition in regions (a) and (b) as well as sensitivity of both parameters. (section 3.1.4) | the area of the computational grid. In general, the tsunami deposits are greatly affected by local micro-topography (Sugawara et al., 2014a; Jaffe et al., 2016), and it is difficult to fit the modelled layer thickness with the observed layer thickness using DEM averaged in a computational grid. Therefore, we introduce the concept of cumulative sedimentation, and evaluated the scale of the amount of sediment movement generated. Although the modelled layer thickness typically overestimates the observed layer thickness by +7%, such low variation suggests a relatively successful reproduction of the observed dataset (Figure 7).

Page 18 Line 454-463
Figure 9 shows the topographical changes and thickness of sediment layer in this calculation for each grain size, and Table 3 shows the volume of erosion and deposition in regions (a) and (b). These evidences show that smaller the particle size is, the greater the topographic change. This can be understood by the smaller the particle size, the larger the Shields number in Eq. (10), which indicates the ease of sediment transport, and the greater the amount of bed load in Eq. (8). However, Figure 9 suggested that the qualitative characteristics of sediment transport are same in the three cases, due to the local erosion position of the beach in region (a) and (b) did not change for any particle size. And then, comparing the tsunami sediment thickness in Figure 10 the errors of the cumulative volume of d = 0.314 mm and d = 0.285 mm are -63% and -55%. Therefore, the grain size of d = 0.127 mm is considered to show the better reproducibility.

Page18 Line 464-474
Figure 11 shows the topographical changes and thickness of sediment layer in this calculation for each bottom roughness coefficient, and |

| | | |
|---|---|---|
| | | Table 4 shows the volume of erosion and deposition in regions (a) and (b). These evidences show that larger the value of roughness coefficient $n\_s$ is, the greater the topographic change. This can be understood by larger the roughness, the larger the Shields number in Eq. (10) because the friction velocity is proportionate to $n\_s$. Therefore, an increase in the roughness coefficient indicates the ease of sediment transport, and the greater the amount of bed load in Eq. (8). However, Figure 11 suggested that the qualitative characteristics of sediment transport are same in the three cases, due to the local erosion position of the beach in region (a) and (b) did not change for any bottom conditions. And then, comparing the tsunami sediment thickness in Figure 12, the errors of the cumulative volume of $n\_s= 0.025$ s/m1/3 and $n\_s= 0.035$ s/m1/3 are -8% and 13%. Therefore, the roughness coefficient of $n\_s= 0.030$ s/m1/3 is considered to show the better reproducibility. |
| The figure quality needs to be improved, at least make sure the fontsize is consistent in all figures, not extra large (Figure 7-9) or extra small (Figure 10). Pay attention to the captain of each figure, make sure they are consistent with the legends inside the figure (see Figure 7 and Figure 8). | We have revised and improved quality of the mentioned figures. | Please see all figures |

**Dear Editor,**

Thank you for your time and sending us your decision. We have made corrections to both reviewers as shown below. Corrections made based on suggestions are shown in red.

**Reply to reviewer no. 3**

We highly appreciate the time spent for the review comments from the reviewer especially those minor corrections (our type errors) and pointed out many points that clarifications are needed. We are happy that the reviewer is happy and highly evaluated our manuscript. Please find our responses and corrections as shown below.

| Reviewer comments | Our answers | Corrected manuscript |
|---|---|---|
| - All the manuscript: In fact, in the abstract the authors conclude that "Our modelling approach confirms that beaches on Phra Thong Island were significantly eroded by the 2004 tsunami" but the analysis of the results, as displayed in figure 5, also show a lot of shoreline accretion. In fact, in most locations' shoreline seems to have experienced a minor accretion (this is especially clear in figure 5a) while significant erosion is only observed at localized sections of the coast. In fact, the large longshore variability remains mostly unresolved, an should be further discussed in the manuscript. | We agreed that in general (as the whole island) there is slightly deposition along the shoreline in Fig. 5. As shown in Fig. 13 (now as Fig.15), where you can see that most deposition is located in shallow sea area. However, we can also see largely and locally eroded areas as shown in Fig. 6 that our model can accurately reproduce. Discussion of the deposition in general of the whole island is actually been addressed in the first and second paragraph of section 4.1. | Please see Page 1 Line30-31 …Our modelling approach suggests that beaches located in two regions on Phra Thong Island were significantly eroded by the 2004 tsunami… Please see Page 5 Figure 2 for locations of the two regions. Caption: …Dashed squares are the beach where erosion was confirmed from satellite image. Please see Section 4.1 Line 574-597 "On Phra Thong Island, the 2004 IOT wave was large enough to expose the nearshore sediments and entrained most of its sediments from the shallow offshore region (below 5m). The wave ran up the exposed nearshore area while retaining sediment from the shallow offshore region. The sediment concentration gradually increases as the wave runs up the relatively long distance of the exposed nearshore zone, and became sediment-saturated as the wave reached the shoreline, making it difficult for new sediment to be eroded further. This explains why there was little erosion of the beach during the inflowing wave, and may be a characteristic sediment transport properties of shallow beaches like those on Phra Thong Island. The numerical simulation results suggest that there is little transportation of sediments from beach by the first inflowing wave and that inland tsunami deposits |

| | | |
|---|---|---|
| | | originated from the nearshore environment. This finding validates Sawai et al. (2009)'s observation that the 2004 IOT entrained diatoms from shallow offshore waters at Phra Thong Island, and Pham et al. (2018)'s observation that sediment grain sizes and mineralogy were most similar to those of nearshore sediments. Figure 15 shows the results of the calculated sediment deposition both onshore and offshore Phra Thong Island. From the modelling results, most of the eroded sediment was deposited in shallow nearshore environments in water less than approximately 5 m deep.

The simulations show that the eroded sediments were deposited in the nearshore zone during backwash (Fig. 15), which primed the coastal zone for rapid coastal recovery. The removal of sediment from the onshore coastal zone also generated accommodation space that may have contributed to the coastal recovery process. Future studies can build on these findings to determine the extent of sediment transport and deposition, and identify the processes of coastal recovery on Phra Thong Island." |
| The first statement of the conclusions "First, it was confirmed by comparing the measured and calculated values of the sediment layer thickness that the location of beach run off identified on Phra Thong Island was reproducible and consistent with sediment transport results", do not seem to have correspondence with the data presented in the paper. | This statement is correspondence with results shown in Figs 6 and 7.
Fig. 6 shows a comparison between our simulated shoreline and the actual shoreline after the tsunami. It can be seen that we can reproduced the same locally eroded areas in both regions a and b.
Fig. 7 shows a comparison of the sediment thickness. It can also be seen that the simulation could reproduced the cumulative volume with 7% error of overestimation. | Please see Page 25 Line 640- Page 26 Line 642

First, it was confirmed by comparing simulated results of the shoreline and sediment layer thickness that the location of beach runoff identified on Phra Thong Island was reproducible and consistent with sediment transport results (Figs. 6 and 7). |
| In section 3.1.2 "change of shoreline" authors refer that sediment transport models confirm the erosion as portrayed by satellite images, but do not present satellite images before and after the tsunami occurrence. | Thank you for the suggestion. We agreed with you that presenting shoreline before the tsunami would help to reader for better understanding the situation. We have added new figures (in Fig. 6) comparing shoreline before and | Please see Page 14 Figure 6 Caption:
…satellite images before and after the tsunami (20 Dec. 2004 and 30 Jan. 2005), which is overlain by the modelled extent of erosion showing that the modelled results |

| | | |
|---|---|---|
| An objective comparison of model performance with satellite date with quantitative error statistics should also be present (e.g. brier skill score). The display of satellite images just before the tsunamic also would help the reader to have perception if the coastal embankments portrayed in image 6 existed before the tsunami. | after the tsunami and pointing locations of serious erosions in regions a and b. However, this is the only satellite image before the tsunami that was available which has low resolution, quantitative comparison is difficult.

We have also added, shoreline from topography data (before the tsunami) for the comparison instead. | closely match the observed changes. The red line is the calculated shoreline after the tsunami, and the yellow line is the shoreline before the tsunami…
…a1) Satellite image before the tsunami in region (a), a2) Satellite image after the tsunami in region (a), b1) Satellite image before the tsunami in region (b), b2) Satellite image after the tsunami in region (b)… |
| The comparison of tsunami deposit thickness (figures 7 and 8) with the observed sediment layer also casts serious doubts on the model performance. In fact, the locations where the larger deposition were found (> 2000 inland) are the locations where the model predicted no accumulation. Moreover, a scatter plot with estimated layer thickness against observed thickness should be presented, supplemented with objective error statistics. Although authors discuss some discrepancies, this section should be expanded. | Previous studies (e.g. Sugawara et al., 2014a; Watanabe et al., 2018) have mentioned that it is difficult to evaluate the simulation results using point-by-point comparison. Instead, they suggested to use "cumulative volume" as a general way to evaluate the total transported sediment. We also calculated this cumulative volume and found that the error between the survey and simulation is only 7%. Therefore, we judged that our simulation has good reproductively.

We have also added more discussions on the overestimation in area less than 1 km and underestimation in area over 2 km. (Please see section 3.1.3 and Fig. 8 in detail) | Page 15 Line 390- Page 16 Line 422
Figure 7 shows a comparison of layer thicknesses at each site (black circles for measured results and white circles for simulated results), which shows that most of the sites are overestimated within 1 km from the shoreline and underestimated at distances greater than 2 km from the shoreline. The model specification and topographical data can be considered as the major causes of this error.
First, considering the overestimation within 1 km of the inundation distance, it is found that the STM has a setting of the maximum suspended concentration, Cmax as 37.7% (Xu, 1999a and 1999b in section 2.4.). The computed suspended concentration in this area is higher than Cmax. Therefore, the surplus sediment is forced to be deposited in this zone causing overestimation. Pham et al. (2018) found that the source of tsunami deposits in Phra Thong Island is mainly the sediment from nearshore. In other words, the first wave, which had the highest wave height, eroded a large amount of sediment in the nearshore and transported a large amount of sediment inland. Therefore, it is considered that the maximum concentration was reached during the first wave run-up because of the very high concentration of suspended sediment, which led to the overestimation of the forced |

sedimentation in the simulation. Second, considering the underestimation of the deposition in inundation distances of 2 km or more, the most likely reason is the computational grid and the model specification. Previous studies have shown that tsunami deposits are highly affected by locality features (e.g. Sugawara et al., 2014a; Watanabe et al., 2018). As shown in three locations with the actual measured deposit thickness (dashed boxes) in Fig. 8, it can be seen that most of the measured thickness is zero which indicate and support the reasons of localized deposition. Although the computational grid is very fine ($\Delta x\_6= 5$ m), it is difficult to reproduce local sedimentation with averaged elevation data. In addition, this STM apply single grain size and can only model sandy deposit. Sugawara et al. (2014a) conducted tsunami sediment transport simulation in the Sendai Plain and discussed on the transportation possibility of finer grained sandy and muddy sediment. Muddy sediments were also found in the Sendai plain at a distance of 2 km or more from the 2011 tsunami. The same STM-based tsunami sediment transport simulation could not be reproduced for such situation which is a limitation of the model for applying to sandy soil and a single grain size. Therefore, it is possible that muddy or very fine-grained sediment was deposited even at the three sites that were underestimated in the simulations which is difficult to represent in the current model.

Page 16 Line 425-432 and Figure 8
…the line of "Cumulative volume" show the cumulative deposition expressed at each point by the sediment thickness multiplied by the area of the computational grid. In general,

| | | the tsunami deposits are greatly affected by local micro-topography (Sugawara et al., 2014a; Jaffe et al., 2016), and it is difficult to fit the modelled layer thickness with the observed layer thickness using DEM averaged in a computational grid. Therefore, we introduced the concept of cumulative sedimentation, and evaluated the scale of the amount of sediment movement generated. Although the modelled layer thickness typically overestimates the observed layer thickness by +7% , such low variation suggests a relatively successful reproduction of the observed dataset (Figure 7). |
|---|---|---|
| When comparing the model results with validation data, it seems that it would be more useful to present more detailed data, even though at a single site. | Similar to our answer to the previous question, we have selected some areas located over 2 km to discuss about the simulation results with validation data. | Page 15 Line 404-Page16 Line 422

Second, considering the underestimation of the deposition in inundation distances of 2 km or more, the most likely reason is the computational grid and the model specification. Previous studies have shown that tsunami deposits are highly affected by locality features (e.g. Sugawara et al., 2014a; Watanabe et al., 2018). As shown in three locations with the actual measured deposit thickness (dashed boxes) in Fig. 8, it can be seen that most of the measured thickness is zero which indicate and support the reasons of localized deposition. Although the computational grid is very fine ($\Delta x\_6= 5$ m), it is difficult to reproduce local sedimentation with averaged elevation data. In addition, this STM apply single grain size and can only model sandy deposit. Sugawara et al. (2014a) conducted tsunami sediment transport simulation in the Sendai Plain and discussed on the transportation possibility of finer grained sandy and muddy sediment. Muddy sediments were also found in the Sendai plain at a distance of 2 km or more from the 2011 tsunami. The same STM- |

| | | based tsunami sediment transport simulation could not be reproduced for such situation which is a limitation of the model for applying to sandy soil and a single grain size. Therefore, it is possible that muddy or very fine-grained sediment was deposited even at the three sites that were underestimated in the simulations which is difficult to represent in the current model. |
|---|---|---|
| Concerning model application, there are lot simplifications that can affect model results that are not properly justified or validated. Sediment transport magnitude and consequent morphological changes are largely dependent on the chosen values for the parameters displayed in table 4. The assumption that some parameters assume a constant should also be justified namely the friction speed (or is this critical friction?) and bottom slope correction factor. | In tsunami sediment transport model, those parameters were often simplified for simulation. Based on previous studies, those parameters were generally given by fixed value which were also used in this study. Parameters for sediment transport simulation are summarized in revised Table 2. Additional equations (equations 11-16) have been added and explained for further clarifications of the parameters. Friction speed is corrected to critical friction. | Please see in detail for the added equations 11-16 and revised Table 2 |
| - Page1 Line 33: how can authors "confirm" if there is no observational data? | Changed "suggests" | Our modelling approach suggests that beaches… |
| - Page 1 Line 73: to support the statement "reproducibility has been confirmed by comparison between the calculated and measured values" a reference is needed. | We have corrected by adding references as suggested. | Please see Page 3 Line 84-85 (e.g. Li et al., 2012; Ranasinghe et al., 2013; Sugawara et al., 2014a; Yamashita et al., 2015; Yamashita et al., 2016) |
| - Page 5 Figure 2: a graphical scale or different gridline numbering should ease a better perception of the scale of the figure. | We have added a scale to each figure | Please see Page 13 Figure 5 |
| - Page 9 Line 234: the use of " Manning's roughness coefficient was fixed at n = 0.025" contradicts the recognition (l438) that "bottom surface roughness greatly affects sediment transport" | We are sorry for such confusion. We agreed with this so that we have added another section on sensitivity analysis of roughness coefficients (ns = 0.025, 0.030 and 0.035). As results shown in section 3.1.4, We have also tested sensitivity analysis for additional two bottom coefficients at ns = 0.025 s/m1/3 and | Page 11 Line 302-314. For the bottom conditions of STM, the roughness coefficient was fixed at ns = 0.030 s/m1/3, and the entire area of Region 6 was considered the movable bed. In general, when simulating tsunami sediment transport, it is necessary to determine the roughness |

ns = 0.035 s/m1/3 which are within the range of commonly used in previous studies (e.g. Sugawara et al., 2014a, b)

From the sensitivity analysis result, it can be seen that smaller grain size causes larger bathymetric change (Table 4) but locations of erosions and bathymetric changes in both regions (a) and (b) still have the same trend. In addition, error of the cumulative volume for ns = 0.030 is the smallest (+7 %) while -8 % and +13 % for ns = 0.025 and ns = 0.035 respectively. Therefore, we judged that using one single roughness coefficient (ns = 0.030) is reliable and suitable for further discussion in later of the paper.

coefficient according to land use. However, since there is no land use map before the tsunami on Phra Thong Island, a fixed value was used, similar to previous studies (e.g. Sugawara et al., 2014a, b; Yamashita et al., 2015; Yamashita et al., 2016). However, Sugawara et al. (2014a) showed that the variation in Manning's roughness coefficient for the sand beds may affect the general distribution pattern of sediment deposits and erosions across the artificial topographic features with much higher roughness coefficient such as artificial canal, road and populated residential areas. Therefore, it is necessary to perform a sensitivity analysis on the roughness coefficient. Phra Thong Island has no such artificial topographic features and using the single roughness coefficient shall give promising results, so we tested sensitivity analysis for two bottom conditions at ns = 0.025 s/m1/3 and ns = 0.035 s/m1/3 which are within the range of commonly used in previous studies(e.g. Sugawara et al., 2014a, b)

Please see Page 18 Line 464-474
Figure 11 shows the topographical changes and thickness of sediment layer in this calculation for each bottom roughness coefficient, and Table 4 shows the volume of erosion and deposition in regions (a) and (b). These figures show that larger the value of roughness coefficient $n_s$ is, the greater the topographic change. This can be understood by larger the roughness, the larger the Shields number in Eq. (10) because the friction velocity is proportionate to $n_s$. Therefore, an increase in the roughness coefficient indicates the ease of sediment transport, and the greater the amount of bed load in Eq. (8). However, Figure 11 suggests that the qualitative characteristics of

| | | |
|---|---|---|
| | | sediment transport are the same in the three cases, due to the local erosion position of the beach in region (a) and (b) did not change for any bottom conditions. And then, comparing the tsunami sediment thickness in Figure 12, the errors of the cumulative volume of $n_s = 0.025$ s/m$^{1/3}$ and $n_s = 0.035$ s/m$^{1/3}$ are -8% and 13%. Therefore, the roughness coefficient of $n_s = 0.030$ s/m$^{1/3}$ is considered to show the better reproducibility. |
| - Page 9 Line 228-239: presents some formatting problems | Corrected | |
| - Page 9 Lines 238: is the "limit Shields" is the critical Shields parameter? The authors should differentiate the Shields parameter from bottom shear stress (eq. 10) | Corrected.We are sorry for the mistyping. | Please see Page 11 Line 317 The critical Shields number … |
| - Page 10 Table 2 - The use of significant figures should be improved | Corrected | Please see Page 6 Table 2 |

**Dear Editor,**

Thank you for your time and sending us your decision. We have made corrections to both reviewers as shown below. Corrections made based on suggestions are shown in red.

**Reply to reviewer no.4**

We highly appreciate the time spent for the review comments from the reviewer especially those minor corrections (our type errors) and pointed out many points that clarifications are needed. We are happy that the reviewer is happy and highly evaluated our manuscript. Please find our responses and corrections as shown below.

| Reviewer comments | Our answers | Corrected manuscript |
|---|---|---|
| - It is not clear how the TUNAMI N2 and STM were coupled. The authors need to provide more detailed information such as conformity of grid size, time step, and bathymetric-topography data. Furthermore, it is not clear whether the bed level in the TUNAMI N2 were also updated after sediment transport or not. | More explanations have been added as below.
- Coupling TUNAMI and STM in section 2.4.2
- Grid size and source of the bathymetry/topography data were in fact mentioned in the original manuscript in section 2.2.
- Simulation time step was actually also mentioned in the original manuscript in section 2.5
The bed level in TUNAMI was also updated as feedback from STM. | Please see Page 8 Line 207-209

For each time step, the STM receives the total flow fluxes from TUNAMI-N2 and calculates the change of seafloor and land surface and feeds this to the next time step of the TUNAMI-N2 model. |
| - The reasons to run the simulation for 6 hours is not clear. Any data show the tsunami propagation at this area lasted in 6 hours? | In fact, we have also performed the simulation for 12 hours. We found that there is no significant different in suspended sediment concentration as well as maximum erosion and deposition comparing to the results from 6-hour simulation. Therefore, 6-hour simulation was used for the reproduction of the 2004 tsunami as well as further sensitivity analysis of the grain size and roughness coefficient.
    As shown in appendix of this answer sheet which shows 12 hour simulation results of water level, bathymetric change and suspended sediment concentration at eight points. It can be seen that (i.e. at point no.1) the suspended sediment concentration is not changed after six hours.
    Therefore, simulation time less than 6 hours is not enough as large change in suspended sediment concentration is still happened during 3-5 hours but no change after 6 hours. Six hours simulation was then selected for further analysis | Please see Page 10 Line 296- Page 11 Line 301.

The simulations were calculated over a 0.05 second increment with a 6-hour period in which the test case with a 12 hour period showed the suspended sediment concentration in the vicinity of the shoreline decreased and stabilized. Therefore, 6-hour simulation was used for the reproduction of the 2004 tsunami as well as further sensitivity analysis of the grain size and roughness coefficient. |

| | including sensitivity analysis to reduce the computational cost. | |
|---|---|---|
| - The manning coefficient was treated uniform. Is the coefficient sensitive to the results? No specific sensitivity analysis was done in this research. | We have tested sensitivity analysis for additional two bottom coefficients at $n_s$ = 0.025 s/m$^{1/3}$ and $n_s$ = 0.035 s/m$^{1/3}$ which are within the range of commonly used in previous studies (e.g. Sugawara et al., 2014a, b)

From the sensitivity analysis result, it can be seen that smaller grain size causes larger bathymetric change (Table 4) but locations of erosions and bathymetric changes in both regions (a) and (b) still have the same trend. In addition, error of the cumulative volume for ns = 0.030 is the smallest (+7 %) while -8 % and +13 % for $n_s$ = 0.025 and $n_s$ = 0.035 respectively. Therefore, we judged that using one single roughness coefficient (ns = 0.030) is reliable and suitable for further discussion in later of the paper. | Page 18 Line 464-474.
Figure 11 shows the topographical changes and thickness of sediment layer in this calculation for each bottom roughness coefficient, and Table 4 shows the volume of erosion and deposition in regions (a) and (b). These figures show that larger the value of roughness coefficient $n_s$ is, the greater the topographic change. This can be understood by larger the roughness, the larger the Shields number in Eq. (10) because the friction velocity is proportionate to $n_s$. Therefore, an increase in the roughness coefficient indicates the ease of sediment transport, and the greater the amount of bed load in Eq. (8). However, Figure 11 suggests that the qualitative characteristics of sediment transport are the same in the three cases, due to the local erosion position of the beach in region (a) and (b) did not change for any bottom conditions. And then, comparing the tsunami sediment thickness in Figure 12, the errors of the cumulative volume of $n_s$= 0.025 s/m$^{1/3}$ and $n_s$= 0.035 s/m$^{1/3}$ are -8% and 13%. Therefore, the roughness coefficient of $n_s$ = 0.030 s/m$^{1/3}$ is considered to show the better reproducibility. |
| This paper also attempts to bring recovery process of the beach, which I do not see where the recovery has taken place. Usually, beach recovery process takes years after a tsunami or storm surges. The impacts of the tsunami was performed by the models, but recovery process of the beach is not. | We believe that there are two steps in order to understand the recovery process after the tsunami, 1) Short term: Sudden bathymetric change at the time/right after the tsunami and 2) Long term: Slowly bathymetric change by wind and wave conditions.
The main objective of this study was 1) which was to investigate characteristic of sediment transport as well as erosional and depositional process affected by the 2004 tsunami using TUNAMI and STM models. One of our main finding is that the sediment deposited in the | Page 3 Line 94-96
The main objective of this study is to investigate the short term conditions of sediment transport such as erosional and depositional process and establishes the baseline sediment conditions that led to further investigation of the long term recovery of the Phra Thong Island coastline after the 2004 IOT.

Page 26 Line 650-652
These erosional and depositional processes demonstrate the locations of sediment removal and |

| | | |
|---|---|---|
| | shallow area near the shoreline which could be a reason for further investigation of recovery process and input for further investigation in 2). Other objectives of this study were that this study will improve understanding of STM in a different coastal area as well as the first step for investigating Palaeo tsunami in the study area. To better clarify our main study objective, some parts of the manuscript were modified accordingly. | subsequent deposition during the different phases of the first tsunami wave on Phra Thong Island which will serve as an important baseline of sediment sources for further study of the recovery process. |
| - Backwash created deposition at the offshore area instead of erosion in other study area. But, this study revealed the opposite. Author needs to review some more cases that could give different result. | We meant that in our study, backwash also created erosion at the offshore area but at the end (in total) we got deposition in offshore area (Fig. 15). In other word, as backwash of the first wave ended, the water still contained a high suspended sediment concentration and this was deposited in the nearshore area. This process is similar to what happened in the case of the 2004 Indian Ocean tsunami and the 2011 Japan tsunami. We have also reviewed and added references accordingly. | Page 23 Line 548-568. Beach erosion due to backwash has also been confirmed in for the 2004 IOT in Sri Lanka and the 2011 Tsunami along the Sendai Plain and at Rikuzentakata. (eg Tanaka et al., 2007, Tanaka et al. 2011, Yamashita et al. 2015, 2016). On the Sendai Plain, the estuary section of the old river tends to increase the return flow due to the tsunami (Tanaka et al., 2007, Tanaka et al. 2011). Therefore, there is a possibility that the region (a) and (b) (Fig. 2, 5 and 6) where local beach erosion of the backwash occurred on Phra Thong Island are the old river part. Conversely, the entire beach was eroded by the return flow in Rikuzentakata (Yamashita et al., 2015, 2016), but no erosion was observed along the entire beach on Phra Thong Island and the Sendai Plain. Yamashita et al. (2015, 2016) suggested that the difference between Rikuzentakata and the Sendai Plain may be related to the horizontal distance of the plains. On the Sendai Plain, the inland topographic gradient is small, the inundation distance is long and the inland inundation depth tends to be small. Therefore, the potential energy that the inundation depth changes to kinetic energy during the backwash (return flow) becomes relatively small. The Sendai Plain and Phra Thong Island are flooded |

| | | |
|---|---|---|
| | | plains over 2 km inland and have similar topographical features.

From the above reasons, the local beach erosion due to the return flow on Phra Thong Island occurred at the mouths of tidal channels and within tidal channels and that minimal erosion occurred across the wider beach ridge strand plain. As backwash of the first wave ended, the water still contained a high suspended sediment concentration and this was deposited in the nearshore environment at less than 5 m water depth (Figure 15). After that, no significant topographic change was found. Thus, this modelling shows that most of the sediment that eroded from the onshore area was deposited in the shallow nearshore zone. |
| - Diffusion coefficient in Equation 6 has different symbol in the paragraph explaining the equation | We have added more explanations to better explain our sediment transport model. Equations 8-9, 11-16 have been added. | Page 9-10. Please see equations 8-16 and related explanations for more detail. |
| - Figure 10, three figures in the last row have no clear explanation: to what time these figures were meant to? Please provide sufficient information and discuss this properly. | We have improved quality of Fig. 10 (now as Fig. 14) as suggested (size, elapsed time and others to improve the visibility). Also more discussions about this result are added in section 3.2.1 | Please see Fig. 14 and Page 21 Line 523-535

It should be noted that, there will be no increase in suspended sediment when the suspended sediment is saturated in the model and is the likely reason that the beach was not eroded by the inflowing first wave. Although there is a possibility that the beach was actually eroded, the numerical results suggest that the erosion in shallow coastal waters (deeper than 5 m but shallower than 10 m) resulted in a very high concentration of suspended sediment when the inflowing first wave entered  -5 m to the beach section of the coast and sediment ceased to be entrained. Pham et al. (2018) found that the source of the 2004 IOT deposits on Phra Thong Island was from the nearshore (depth < 15m). This means that large scale erosion in shallow water has occurred and a large amount of sediment has been transported inland which agrees with the simulation results. |

| | | Therefore, it is highly likely that the sediment concentration was very high when it reached the beach during the 1st inflowing wave. Takahashi (2012) showed that when the suspended sediment is in a high concentration state, turbulence is suppressed and the ability to retain suspended sediment may decrease. Therefore, it is highly probable that the same phenomenon occurred on Phra Thong Island and the beach erosion during the inflowing wave was suppressed. |
|---|---|---|

Appendix

[Figure]

Locations of eight points used in the 12 hour simulation

[Figure]

Simulation results in No.1~No.8
(WL : water level; BL : bottom level; Cs : Concentration of suspend sediment)